# Molecular insights into *Vibrio cholerae*'s intra-amoebal host-pathogen interactions

Charles Van der Henst[1], Audrey Sophie Vanhove[1], Natália Carolina Drebes Dörr[1], Sandrine Stutzmann[1], Candice Stoudmann[1], Stéphanie Clerc[2], Tiziana Scrignari[1], Catherine Maclachlan[2], Graham Knott [2] & Melanie Blokesch [1]

*Vibrio cholerae*, which causes the diarrheal disease cholera, is a species of bacteria commonly found in aquatic habitats. Within such environments, the bacterium must defend itself against predatory protozoan grazers. Amoebae are prominent grazers, with *Acanthamoeba castellanii* being one of the best-studied aquatic amoebae. We previously showed that *V. cholerae* resists digestion by *A. castellanii* and establishes a replication niche within the host's osmoregulatory organelle. In this study, we decipher the molecular mechanisms involved in the maintenance of *V. cholerae*'s intra-amoebal replication niche and its ultimate escape from the succumbed host. We demonstrate that minor virulence features important for disease in mammals, such as extracellular enzymes and flagellum-based motility, have a key role in the replication and transmission of *V. cholerae* in its aqueous environment. This work, therefore, describes new mechanisms that provide the pathogen with a fitness advantage in its primary habitat, which may have contributed to the emergence of these minor virulence factors in the species *V. cholerae*.

[1] Laboratory of Molecular Microbiology, Global Health Institute, School of Life Sciences, Station 19, EPFL-SV-UPBLO, Ecole Polytechnique Fédérale de Lausanne (EPFL), 1015 Lausanne, Switzerland. [2] Bioelectron Microscopy Core Facility (BioEM), School of Life Sciences, Station 19, EPFL-SV-PTBIOEM, Ecole Polytechnique Fédérale de Lausanne (EPFL), 1015 Lausanne, Switzerland. Correspondence and requests for materials should be addressed to M.B. (email: melanie.blokesch@epfl.ch)

The diarrheal disease cholera is not extinct. Seven cholera pandemics have been recorded in modern history and the latest is still ongoing[1,2]. Cholera is caused by ingestion of the bacterium *Vibrio cholerae*. Toxigenic strains of this species are capable of damaging the host due to the presence of so-called virulence factors, which refers "to the elements (i.e., gene products) that enable a microorganism to colonize a host niche where the organism proliferates and causes tissue damage or systemic inflammation"[3]. Bacterial strains without these factors are usually attenuated with respect to the infection process.

For *V. cholerae*, the two major virulence factors, cholera toxin and the toxin-coregulated pilus, have a pivotal role in the infection, but additional minor virulence factors have also been identified. These include factors such as outer membrane porins, a pore-forming hemolysin, diverse proteases, *N*-acetyl-glucosamine binding protein, flagellum-based motility, to name a few[4–7]. The hemagglutinin/protease (HapA), for example, is a zinc-metalloprotease[8], which was first identified due to its mucinase activity[9]. This enzyme was later demonstrated to not only cause hemagglutination but to also hydrolyze fibronectin, mucin, and lactoferrin, all of which were thought to contribute to the host defense against *V. cholerae*[10]. HapA also causes cell rounding, loss of the barrier function of the epithelial layer, and, ultimately, detachment of cells under tissue culture conditions[11,12]. Consistent with these in vitro activities, a tenfold increase in the 50% lethal dose in the absence compared to the presence of the HapA protease for *V. cholerae* strains that otherwise lack cholera toxin was reported[13].

A second well-characterized minor virulence factor that is widespread among *Vibrio* species[14] is hemolysin (HlyA), which is also known as *V. cholerae* cytolysin or vacuolating toxin. Hemolysin is a secreted protein belonging to the family of pore-forming toxins (PFTs), as it forms aqueous channels in host cells in vitro[15]. In vivo data by Ichinose et al.[16] showed that purified hemolysin induced intestinal fluid accumulation in rabbits and in orally inoculated suckling mice, two commonly used animal models of cholera. The authors therefore concluded that hemolysin was an enterotoxic factor that contributed to gastroenteritis caused by non-pandemic *V. cholerae* strains[16]. The role of HlyA as enterotoxic or diarrheagenic factor, especially in pandemic isolate-derived vaccine strains lacking cholera toxin, was later confirmed by Alm et al.[17] who also demonstrated that mice infected with hemolysin-deficient *V. cholerae* survived longer than those inoculated with their hemolysin-positive parental strains. More recent studies using streptomycin-treated adult mice and pandemic isolates of *V. cholerae* showed that hemolysin was the major cause of lethality in this animal model of cholera[18]. Moreover, it has been suggested that HlyA and other secreted accessory toxins modify the host environment, thereby contributing to prolonged disease-free colonization that could contribute to disease transmission via asymptomatic carriers[19].

It was proposed that *hlyA* might be located on a pathogenicity island due to the close proximity of additional genes that likewise encode extracellular enzymes[20]. This region encodes apart from hemolysin (*hlyA*) a lipase (*lipA*), a metalloprotease (*prtV*), and a lecithinase/phospholipase (*lec*; also known as thermolabile hemolysin). The lecithinase activity of the latter enzyme was characterized[21,22], but molecular details and its concrete function are still unknown.

To date, a limited number of studies have shown that minor virulence factors, such as those mentioned above, are also advantageous in an environmental context[23,24]. Indeed, despite *V. cholerae* being an aquatic bacterium that is well adapted for survival in this environment, the molecular details about *V. cholerae*'s environmental lifestyle are still lacking. To address this knowledge gap, we aimed to elucidate the molecular mechanisms that *V. cholerae* would use as part of its environmental lifestyle, such as its interactions with grazing amoebal predators. In this context, we recently demonstrated that *V. cholerae* survives predation by *Acanthamoeba castellanii*, a free-living amoebal species[25]. This amoeba is frequently encountered in aquatic reservoirs and follows a bi-phasic lifecycle: the motile trophozoite form, which grazes on bacterial prey, and the double-walled resistant cyst stage, which forms under harsh conditions[26]. We previously showed that *V. cholerae* interacts with *A. castellanii* through the evolution of two distinct phenotypes. Firstly, upon phagocytosis by the feeding amoebal trophozoite, a fraction of the ingested *V. cholerae* population can resist amoebal digestion, then exit the phagosomal pathway by exocytosis. Notably and in contrast to a previous study[27], we never observed free *V. cholerae* cells in the cytosol of such trophozoites[25], indicating that the bacteria are not released from food vacuoles intracellularly. Secondly, the pathogen colonizes the amoeba's contractile vacuole (CV), which is its osmoregulatory organelle. This organelle expands and contracts in a rhythmic manner based on the accumulation and expulsion of excess water from the cytosol of the cell[28]. While there is still a significant knowledge gap with respect to the detailed mechanisms underlying its osmoregulatory function, it is currently thought that water enters the CV due to a ~1.5 times higher osmolarity of the CV fluid compared to that of the cytosol[29]. The water discharge process, on the other hand, is based on the fusion of the CV and the plasma membrane, a process that is actively driven by myosin-I[30]. This discharge process is essential, as interference with the water expulsion leads to an overfilling of the CV and, ultimately, amoebal cell lysis[30]. With respect to *V. cholerae*, we recently showed that the bacterium enters this organelle through vacuolar fusion of *V. cholerae*-containing food vacuoles with the CV. This creates a replication niche for the pathogen in which *V. cholerae* is maintained even upon amoebal encystment, before it ultimately lyses the host[25]. However, the underlying molecular mechanisms of this host–pathogen interaction remained elusive and were therefore addressed in the current study. We show that *V. cholerae* fine-tunes extracellular enzymes to avoid premature intoxication of its host, allowing the bacterium to take full advantage of the intra-amoebal replication niche for undisturbed and non-competitive proliferation before finally lysing its amoebal host. We also demonstrate that flagellum-based motility is of importance for *V. cholerae* to ultimately escape the ruptured host niche to return to the aquatic environment.

## Results

**Visualization of colonized contractile vacuoles**. Using confocal laser scanning microscopy, we recently demonstrated that *V. cholerae* could enter the CV of *A. castellanii* and replicate within this niche[25], though underlying molecular mechanisms of this host–pathogen interaction remained largely unknown. This is partially due to the low throughput of transmission electron microscopy (TEM), which only allowed for the imaging of hyper-colonizing *V. cholerae* mutant strains, due to their lack of quorum-sensing, while spatial information on wild-type (WT) bacteria within the amoebal vacuoles remained missing[25]. To circumvent this problem, we developed a correlative light and electron microscopy (CLEM) imaging strategy, which allowed us to study this specific host–pathogen interaction for WT *V. cholerae* for the first time at high resolution. We identified *V. cholerae*-infected *A. castellanii* cells by pathogen-produced green fluorescent protein GFP and then targeted these cells for TEM. Using this method allowed us get information on the morphology of the colonizing bacteria and their spatial distribution. As shown in Fig. 1, the morphologies of such intra-vacuolar WT bacteria possessed the

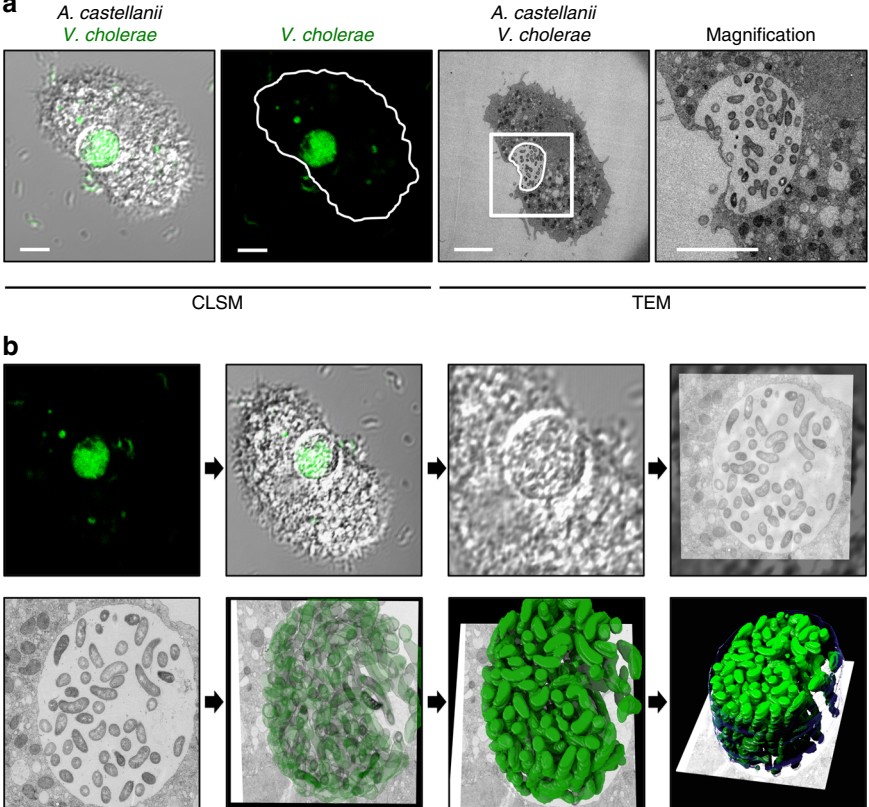

**Fig. 1** Correlative light and electron microscopy (CLEM) for visualizing wild-type *V. cholerae* inside the contractile vacuole (CV). **a** Low- and high-resolution imaging of an infected amoeba. GFP-tagged *V. cholerae* were seen to be localized inside of a CV of *A. castellanii* using confocal laser scanning microscopy (CLSM; low resolution) in fixed samples. Shown are a merged image of the transmitted light channel and the green channel (left) and the green channel image alone (second from left). After staining of the sample, the same amoeba was imaged at high resolution using transmission electron microscopy (TEM; right images). Scale bar in all images: 5 μm. **b** 3D reconstruction of the colonized CV. The region containing the amoeba shown in **a** was serially thin sectioned (50 nm thickness) and serial images were taken with the TEM. These images were then aligned to generate a 3D model of the colonized amoeba. Shown are snapshots of the resulting 3D reconstruction movie (Supplementary Movie 1)

typical *Vibrio* shape. Moreover, the membrane surrounding the CV was largely intact, which was consistent with the non-encysted phenotype of the amoeba (Fig. 1a).

We estimated the number of objects—in this case, bacteria—in the CV using an unbiased stereological approach[31]. To do this, a representative WT *V. cholerae*-colonized amoeba, at 20 h post-primary contact (p.p.c.), was identified in the fluorescent channel. After sample preparation and imaging, the number of bacteria in the vacuole was estimated to be 194 (±60). This number, and the apparent density of bacteria in the 3D reconstruction (Fig. 1b and Supplementary Movie 1), correlates well with the active growth of *V. cholerae* in this organelle, as the vacuole's initial colonization only involved one or a few bacteria[25].

**Absence of HapA leads to aberrant amoebal morphologies**. As the 3D reconstruction suggested that *V. cholerae* cells were densely packed within the vacuole at this later point of the infection, we wondered how the bacteria eventually escaped the vacuole after amoebal encystation and how the timing was properly regulated. To answer these questions, we first took an educated guess approach, which has its limitation but turned out as useful in this study. We therefore tested diverse knockout mutants of *V. cholerae* for their intra-amoebal behavior. We were especially interested in strains that lacked certain extracellular enzymes, as some of those were known to be important in the pathogen's intra-human lifestyle and its transmission to new hosts.

We challenged *A. castellanii* with a HapA-deficient *V. cholerae* strain (Supplementary Table 1) and compared the amoebal response to the WT-challenged condition. By doing this, we observed that a vast majority (79.8%) of infected amoebal cells showed an aberrant morphology upon co-incubation with the *hapA*-minus strain for 20 h, which occurred significantly less often for WT-infected amoebae (11.6%, Fig. 2a, b). These morphological abnormalities ranged from a shrinking or compacting phenotype towards pseudopodia retraction and detachment, all of which ultimately abolished amoebal grazing (Fig. 2 and Supplementary Fig. 1). Complementation assays, involving a genetically engineered derivative of the *hapA*-minus strain with a new copy of *hapA* on the large chromosome (Supplementary Fig. 2), were performed using the amoebal infection protocol and fully restored the WT-infected amoebal morphologies (Fig. 2a, b).

Seeing these striking differences between WT-infected and *hapA*-minus-infected amoebae, we wondered whether the latter might cause "amoebal constipation", meaning the accumulation of undigested phagosomal *V. cholerae* without efficient exocytosis. Indeed, based on the confocal microscopy images, we could localize the accumulated bacteria due to their green fluorescence. However, recognition of the CV in the transmitted light channel was often difficult for *hapA*-minus mutant-infected amoebae due to their compaction and malformation (Fig. 2a). Therefore, it seemed possible that the bacteria were contained in digestive food vacuoles of the endosomal pathway, blocking amoebal digestion

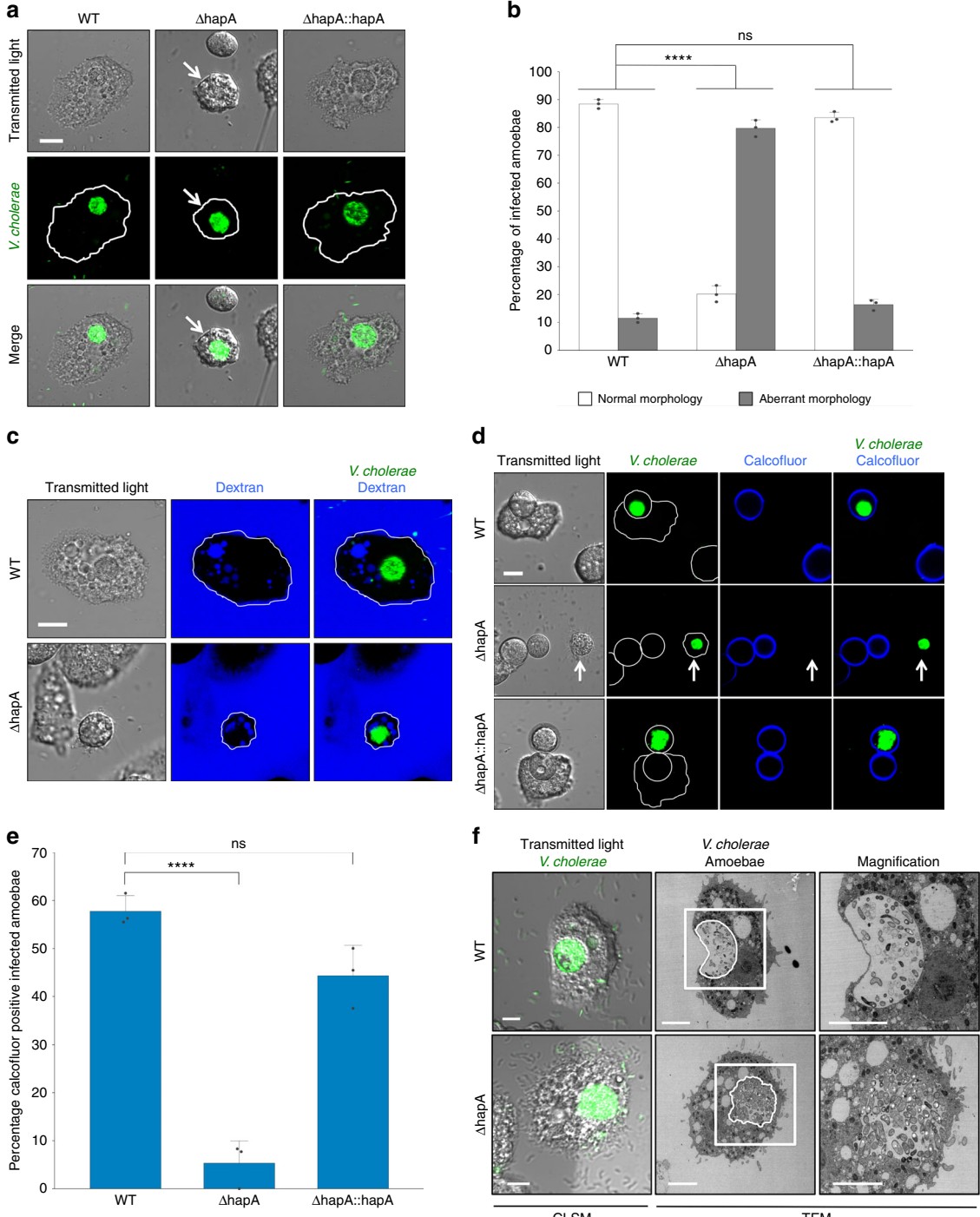

**Fig. 2** Hemagglutinin protease-deficient *V. cholerae* strains intoxicate amoebal trophozoites. **a** CLSM images of amoebae that were colonized by GFP-tagged wild-type (WT), *hapA*-minus (ΔhapA), or the *hapA* complemented (ΔhapA::hapA) *V. cholerae* strains at 20 h p.p.c. Shown are the transmitted light channel (top), the green channel (middle), and the merged image of both channels (bottom). The white arrows highlight the aberrant morphology of the intoxicated amoeba. **b** Quantification of normal and aberrant morphologies of amoeba infected with any of the three bacterial strains mentioned in **a**. **c** Protease-deficient *V. cholerae* mutants do not co-localize with dextran-labeled digestive vacuoles. Labeling of the endosomal pathway using fluorescent dextran shows the residence of WT (top) and ΔhapA (bottom) strains inside the dextran-negative CV. **d** Cellulose is not deposited around amoebae infected with *hapA*-minus bacteria at 30 h p.p.c. Amoebae infected with WT, ΔhapA, or the *hapA* complemented strain (ΔhapA::hapA) were stained with calcofluor to visualize deposited cellulose around the cysts. The white arrows point towards a cellulose-deficient *A. castellanii* infected by the *hapA*-minus strain. **e** Quantification of the calcofluor-stained infected amoebae. Bacterial strains correspond to those in **d**. **f** CLEM images of WT- and *hapA*-minus-infected amoebal trophozoites (details as in Fig. 1). Scale bars: 10 μm (**a**, **c**, **d**) and 5 μm (**f**). All graphs show the averages of three independent biological replicates (±s.d., as shown by the error bars). Statistics are based on a one-way ANOVA. ***$p \leq 0.001$; ns (not significant), $p > 0.05$

as opposed to being localized to the CV, as is the case for WT *V. cholerae*. To distinguish between both scenarios (localization within a digestive vacuole or within the CV), we labeled the amoebal endosomal pathway with fluorescently labeled dextran. As the dextran- and bacteria-derived fluorescent signals did not overlap, these experiments confirmed the intra-CV localization of the *hapA*-minus strain (Fig. 2c).

**HapA-deficiency impairs encystation**. Next, we wondered whether, despite these unusual amoebal morphologies, the life-cycle between *hapA*-minus *V. cholerae* and *A. castellanii* would proceed as previously reported for the WT[25]. Such a progression included amoebal encystation followed by the escape of *V. cholerae* from the CV into the cyst's cytosol, leading, ultimately, to lysis of the amoebal host. To determine this, we imaged infected amoebae at a later time point (30 h p.p.c.) to allow the lifecycle progression to develop. To better distinguish aberrantly rounded cells from cysts, we used calcofluor staining, as this fluorescence stain is known to bind to the deposited cellulose cell wall layer in *Acanthamoeba* cysts[32]. As shown in Fig. 2d (and quantified in Fig. 2e), the ability to form cellulose-positive cysts was severely impaired in amoebae infected with the *hapA*-minus mutant strain. This impaired encystation contrasted greatly to that of both the WT-infected *A. castellanii* and the complemented *hapA*-minus strain (Fig. 2d, e). We therefore concluded that the absence of the HapA protease leads to premature amoebal intoxication and, consequently, a defect in the encystation process.

To better understand the defect in the encystation process, we again used CLEM to obtain high-resolution images of colonized amoebae. The WT-infected CVs seemed to maintain their internal pressure, leading to a clear separation between the content of the CV and the amoebal cytosol (Figs. 1a and 2f), though this evident distinction was lacking in host cells that were infected by the mutant *V. cholerae* strain, which showed signs of impaired membrane integrity (Fig. 2f).

**Uncontrolled hemolysis leads to premature intoxication**. We speculated that the seemingly collapsed CV membranes might have been due to amoebal membrane rupture caused by the *hapA*-minus mutant strain. Given that bacterial pathogens often cause membrane rupture through the secretion of PFTs[33], we wondered whether the secreted hemolysin of *V. cholerae* (HlyA) might be involved in the observed amoebal intoxication. Consistent with this idea was a study by Tsou and Zhu[34] that showed that the HapA protease degrades HlyA in an in vitro assay. This led us to hypothesize that HapA protease-deficient strains of *V. cholerae* would display enhanced hemolysis, which, ultimately, could cause the observed amoebal intoxication. To test this hypothesis, we generated several *V. cholerae* strains that lacked *hapA*, *hlyA*, or both genes simultaneously (Supplementary Table 1) and tested them for hemolysis and proteolysis on blood and milk agar plates, respectively. As shown in Supplementary Fig. 2, there was a strong increase in hemolysis in strains lacking *hapA*, while the absence of *hlyA* fully abolished this activity. Complementation assays, in which the mutant strain contained a new copy of the missing gene elsewhere on the chromosome (Supplementary Table 1), restored the system to that of the WT (Supplementary Fig. 2a and b). Interestingly and consistent with the extracellular localization of both enzymes, we showed that the secreted HapA protease from a WT strain was also able to inactivate HlyA that was released by a co-cultured *hapA*-minus mutant strain. However, when we engineered an *hlyA*-over-expression strain (Supplementary Table 1), the protease activity exerted by the strain itself or from the co-cultured WT bacteria was insufficient to abolish hemolysis (Supplementary Fig. 2).

With these newly constructed strains in hand, we then tested their effect on *A. castellanii*. Consistent with our hypothesis that a *hapA* mutant of *V. cholerae* would possess enhanced hemolysin activity, which, ultimately, would result in amoebal intoxication, we found that the absence of *hlyA* in the protease-minus background restored normal host morphology and cyst formation (Fig. 3a, b and Supplementary Fig. 3a–d). Infection of *A. castellanii* by the *V. cholerae* double-mutant (ΔhapAΔhlyA) also resulted in an intact integrity of the CV membrane (Fig. 3c and Supplementary Fig. 3e). Overproduction of HlyA in a WT background strain, however, fully phenocopied or even aggravated the protease-minus phenotypes (Fig. 3a, b and Supplementary Fig. 3a–d). These data, therefore, support the notion that HlyA is causing amoebal intoxication and that the HapA protease counteracts this effect.

To determine whether HlyA impacts the amoebae through activity inside of the CV instead of in the surrounding medium, we co-infected *A. castellanii* with WT *V. cholerae* (labeled with dsRed) and either a second WT strain as a control or, alternatively, the *hapA*-minus strain or the HlyA-overexpression strain at a 1:1 ratio (all labeled with GFP). We then imaged the amoebae and quantified the number of colonized ones that showed either a normal or aberrant morphology (Supplementary Fig. 4). This experiment suggested that the secreted protease in the WT strain rescues the amoebae from intoxication by protease-minus and, therefore, HlyA-producing *V. cholerae* if both strains occurred within the same CV (Supplementary Fig. 4b and Supplementary Table 2). This rescuing phenotype was not observed when the second strain was the HlyA-overproducing strain (Supplementary Fig. 4c and Supplementary Table 2). Amoebae containing mono-colonized CVs showed the same phenotype as we had observed in the single strain infection experiments (Supplementary Fig. 4 compared to Figs. 2 and 3). These in vivo observations are, therefore, fully consistent with the in vitro hemolysin activity described above and suggest that both enzymes are present and active inside the amoebal CV.

To test whether these HapA-related and HlyA-related phenotypes are conserved in *V. cholerae* strains, we repeated the above-described experiments with diverse pandemic and non-pandemic *V. cholerae* isolates. First, we compared the well-studied pandemic O1 El Tor isolate A1552[35], which we primarily use in our laboratory, with two other O1 El Tor strains, namely C6706 and N16961 (Supplementary Table 1). Strain A1552 was isolated in California from a traveller returning from South America[36], which links this strain to the Peruvian cholera outbreak in 1990s. *V. cholerae* strain C6706 was isolated in Peru in 1991. Notably, as we recently showed that a laboratory-domesticated version of this strain with a severe quorum-sensing defect has spread amongst *V. cholerae* laboratories around the globe[37], we decided to work with an old stock of strain C6706 (named C6706-original in this study; gift from J. Mekalanos, Harvard). We also included the first sequenced *V. cholerae* strain N16961[38], which was isolated in Bangladesh in 1975. This strain carries a frameshift mutation in the gene encoding the major regulator of quorum sensing HapR[39], which we genetically repaired (named N16961-hapR[Rep] in this study; Supplementary Table 1). As shown in Supplementary Fig. 5, all three O1 El Tor pandemic isolates behaved similarly in our amoebal infection assay and confirmed the hemolysin-mediated toxicity of the *hapA*-minus variants. We also extended our analysis to the non-pandemic *V. cholerae* strain Sa5Y, an isolate from the California coast[40]. Interestingly, this strain exerted enhanced hemolysis in vitro compared to the pandemic isolates (Supplementary Fig. 5a) and therefore phenocopied the *hapA*-minus mutants of the pandemic isolates with respect to its toxicity towards *A.*

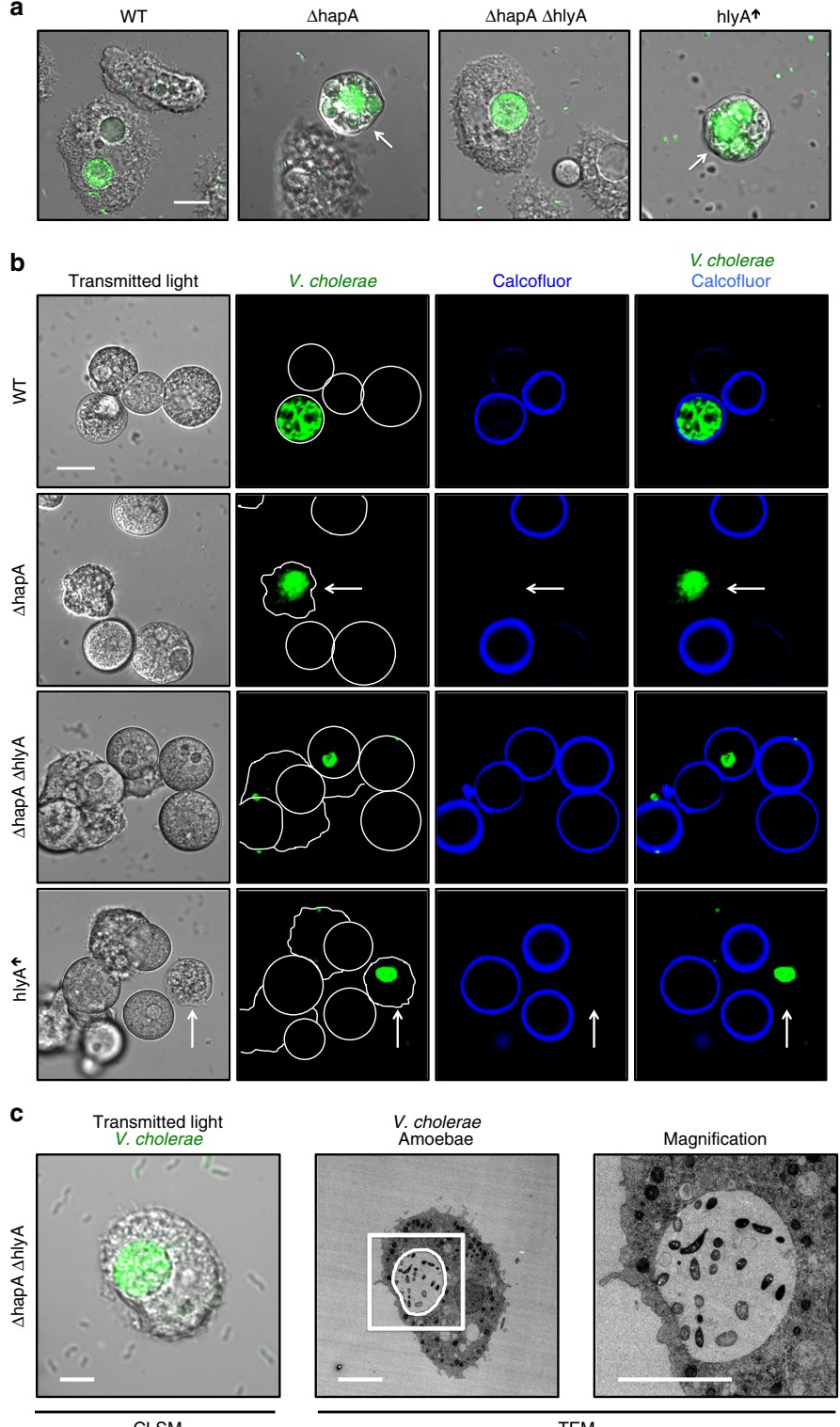

**Fig. 3** Amoebal intoxication in hemagglutinin protease-deficient strains is caused by the PFT hemolysin (HlyA). **a** Aberrant amoebal morphotypes are caused by hemolysin (HlyA). Deletion of the hemolysin gene (*hlyA*) in the protease-minus *V. cholerae* strain (ΔhapAΔhlyA) nullifies intoxication of the amoebal trophozoites upon infection, while the latter is elicited through the overexpression of *hlyA* in the WT background (hlyA↑). All of the bacterial strains were GFP-tagged. Merged images of the transmitted light channel and the green channel were taken at 20 h p.p.c. Aberrantly shaped amoebae are indicated by white arrows. Scale bar: 10 μm **b** Absence of amoebal cellulose deposition is caused by hemolysin. *A. castellanii* cells were infected by the same *V. cholerae* strains as in **a** and, after staining with calcofluor, imaged at 30 h p.p.c. White arrows depict colonized amoebae that are calcofluor-negative and aberrantly shaped. Scale bar: 10 μm. **c** CLEM image of an amoeba that is infected by the protease-deficient and hemolysin-deficient *V. cholerae* strain

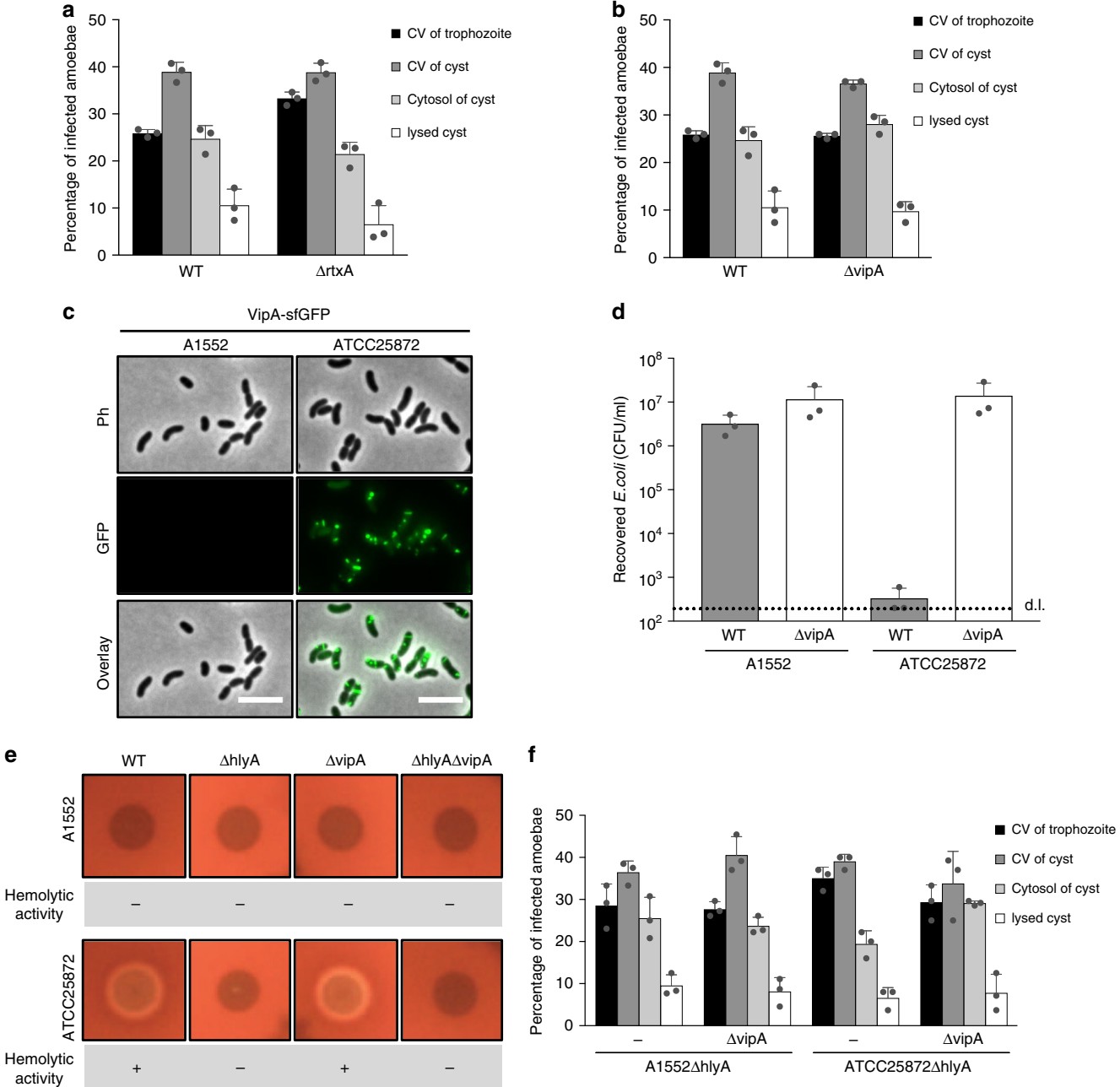

**Fig. 4** The pore-forming toxins RTX and VasX do not intoxicate *A. castellanii*. Distribution of *V. cholerae* wild-type (WT) and a *rtxA*-deficient mutant strain (**a**) or WT and a strain with an impaired type VI secretion system (T6SS) (ΔvipA; deficient for VasX secretion) (**b**) within infected *A. castellanii* at 20 h p.p.c. The considered compartments were the CV of trophozoites, the CV of cysts, and the cytosol of cysts before and after lysis. Values represent averages from three independent experiments (±s.d., as shown by the error bars). **c**, **d** The type VI secretion system of toxigenic strain ATCC25872 is constitutively on. **c** *V. cholerae* strains A1552 (O1 El Tor; pandemic) and ATCC25872 (O37; non-pandemic) carrying a gene that encodes a translational fusion between the T6SS sheath protein VipA and sfGFP (*vipA-sfgfp*) at the native locus of *vipA* were grown for 3.5 h in LB. The images depict cells visualized in the phase contrast (Ph), in the green (GFP) channel, and an overlay of both channels. Scale bar: 5 μm. **d** The constitutive T6SS activity of strain ATCC25872 was confirmed in an interspecies killing assay in which *E. coli* served as prey. Prey survival was assessed by counts of colony-forming units (CFUs) after plating on selective medium. Values represent average from three independent experiments (±s.d.). **e** Hemolysin-based hemolytic activity might mask effects of the T6SS in ATCC25872. *V. cholerae* A1552 and ATCC25872 as well as their ΔhlyA, ΔvipA, and ΔhlyAΔvipA variants were tested for hemolytic activity on blood agar plates. **f** Mutant *V. cholerae* A1552 and ATCC25872 strains lacking hemolysin (ΔhlyA) or hemolysin and the T6SS core protein VipA (ΔhlyAΔvipA) were assessed for their intra-amoebal localization at 20 h p.p.c. as described for **a**. Values are averages from three independent biological experiments

*castellanii* (Supplementary Fig. 5c). Consistent with what we described above for the pandemic isolates, this toxicity towards the amoeba was almost completely abrogated in hemolysin-deficient mutant derivatives (Supplementary Fig. 5c).

**Other PFTs and the type VI secretion system do not harm *A. castellanii*.** As the experiments above suggested that the PFT hemolysin intoxicates amoebal trophozoites, we wondered if two other PFTs of *V. cholerae*, RTX and VasX, would result in similar

outcomes. RTX is a >4000 amino acid protein, which belongs to the family of multifunctional-autoprocessing repeats-in-toxin (MARTX) toxins. Pore formation by RTX serves to translocate effector domains of this large protein across the plasma membrane of eukaryotic cells[41]. Notably, an earlier study suggested that the RTX protein of *V. cholerae* does not impair the membrane integrity of eukaryotic cells (e.g., HEp-2 cells)[42], while a more recent study showed that the RTX toxin of *Vibrio vulnificus* caused pores in human erythrocytes[43]. To test whether RTX contributes to *V. cholerae*'s lifecycle within *A. castellanii*, we engineered an *rtxA*-minus strain of *V. cholerae* (Supplementary Table 1) and compared its behavior to the WT control. As shown in Fig. 4a, no difference in bacterial localization was observed when these two strains were compared, indicating that RTX is not required for *V. cholerae* to establish its replication niche within the amoeba's CV.

The third PFT of *V. cholerae*, VasX, is transported into eukaryotic and prokaryotic cells by the type VI secretion system (T6SS)[44–46]. The T6SS is a multi-protein complex that delivers effector proteins into adjacent cells in a contact-dependent manner[47,48]. Interestingly, the T6SS was first discovered in a non-pandemic strain of *V. cholerae*, strain V52, due to its toxic effect towards the soil amoeba *Dictyostelium discoideum*[49]. To test whether VasX or any other T6SS effector protein (such as the lipase TseL or the actin-cross-linking domain of VgrG1[44–46]) was involved in *V. cholerae*'s interaction with *A. castellanii*, we tested a T6SS-defective bacterial mutant that lacked a core structural component of the secretion system (A1552ΔvipA; Supplementary Table 1) in the amoebal infection assay. No change in the bacterial lifecycle was observed for this mutant when compared to its parental WT strain (Fig. 4b).

Notably, pandemic isolates of *V. cholerae* are known to be T6SS-silent under laboratory conditions[49] and instead require specific environmental cues for T6SS induction[50,51]. We therefore wondered whether the absence of induction would mask potential effects that T6SS-delivered effector proteins could have on *A. castellanii*. To address this question, we took advantage of *V. cholerae* strain ATCC25872 (Supplementary Table 1), a toxigenic O37 serogroup strain that is closely related to strain V52 but in contrast to the latter[52] quorum sensing proficient[53]. First, we confirmed its constitutive T6SS production by imaging a derivative of this strain that carries a translational fusion between the T6SS sheath protein VipA and super-folder GFP[50,54] (Fig. 4c). Constitutive T6SS activity was further demonstrated in an interbacterial predation assay in which strain ATCC25872, but not its *vipA*-negative derivative, efficiently killed co-cultured *Escherichia coli* cells (Fig. 4d). Next, we aimed at challenging amoebae with this T6SS-active strain. However, strain ATCC25872 produced high levels of hemolysin (Fig. 4e), consistent with what was previously shown for strain V52[49]. To avoid that these high levels of HlyA would mask any T6SS-mediated effects due to premature intoxication of the amoebae, we engineered T6SS-proficient or T6SS-deficient mutants in an *hlyA*-minus background. As shown in Fig. 4f, no difference was observed between T6SS-active and T6SS inactive strains in the amoebal infection model or when strain ATCC25872 was compared to the same genetically engineered mutants of pandemic strain A1552. We therefore concluded that neither the RTX toxin nor the T6SS contribute to *V. cholerae*'s interaction with *A. castellanii* under the tested conditions.

**V. cholerae uses its lecithinase to lyse amoebal cysts**. While the experiments described above highlighted the essential nature of the HapA protease for reducing the cytotoxicity of *V. cholerae* towards its amoebal host at early time points, it remained to be

discovered how the pathogen eventually escaped from its amoebal host to return to the environment. We reasoned that the pathogen would need to disrupt the host's plasma membrane. The plasma membrane of *A. castellanii* contains lecithin, which is a mixture of glycerophospholipids that includes a high percentage of phosphatidylethanolamine and phosphatidylcholine[55]. We therefore speculated that *V. cholerae* might use its lecithinase[22] to disrupt this membrane. We generated a *lec*-minus strain and confirmed its impaired lecithinase activity on egg yolk plates (Supplementary Fig. 6). We also complemented the *lec*-minus strain by placing a new copy of *lec* that was preceded by its native promoter onto the chromosome (Δlec::lec). The same genetic constructs were also added into the WT background to generate a *lec*-merodiploid strain of *V. cholerae* (WT::lec). All of the strains were tested for their lecithinase activity in vitro and behaved as expected (Supplementary Fig. 6).

Next, we infected *A. castellanii* with this set of genetically engineered strains. We observed that while the *lec*-deficient strains were able to colonize the CV and escape from it in the cyst stage, the bacteria were unable to ultimately lyse the amoebal host, as visualized through the exclusion of extracellular dextran (Fig. 5). The complemented strain had its cyst lysis capability restored, a phenotype that was enhanced for the *lec*-merodiploid bacteria (Fig. 5). We therefore concluded that the lecithinase enzyme indeed targets and permeabilizes the plasma membrane of the host, thereby triggering the death of the cyst, a phenotype that was conserved in other pandemic and non-pandemic *V. cholerae* strains (Supplementary Fig. 6c and d).

**V. cholerae escapes through flagellum-based motility**. As we observed that *V. cholerae* is highly motile within the amoebal host (Supplementary Movie 2), we wondered whether this motility contributed to the pathogen's intra-amoebal lifestyle and its escape from the succumbed host. We therefore generated non-motile mutants of *V. cholerae* by deleting either the gene that encodes the major flagellin subunit FlaA or the flagellar motor protein PomB (Supplementary Table 1). While the first approach resulted in non-flagellated bacteria, the latter approach led to rotation-deficient but fully flagellated bacteria (Fig. 6a). The motility deficiency of the mutants was further confirmed in in vitro motility assays (Supplementary Fig. 7). Next, we infected *A. castellanii* with these mutant strains. While both mutant strains still infected the amoebal CV, the strains were more static within this niche (Fig. 6b and Supplementary Movies 3 and 4). We therefore wondered whether motility would have a role in the escape of *V. cholerae* from the lysed CV and lysed cysts. To test this idea, we infected amoebae with the WT and the *flaA*-minus strains, which we labeled with different fluorescent proteins, namely dsRed and GFP, respectively. We then took time-lapse movies over several hours to observe the colonizing bacteria of those amoebae that contained both strains within the same CV (Fig. 6c and Supplementary Movie 5). These experiments showed that, compared to the WT strain, the non-motile mutant was retained in the lysed CV (Supplementary Movie 6) and its escape back to the environment was severely impaired after the final lysis of the cyst (Supplementary Movie 7). We concluded, therefore, that motility has a major role in *V. cholerae*'s interaction with *A. castellanii*.

**Discussion**

*V. cholerae*, the bacterial agent responsible for cholera, still poses a global threat to human health. However, apart from its chitin-induced phenotypes, which include chitin catabolism, interbacterial competition, and horizontal gene transfer[50] (reviewed by refs.[56–59]), we know very little about its environmental lifestyle and its potential

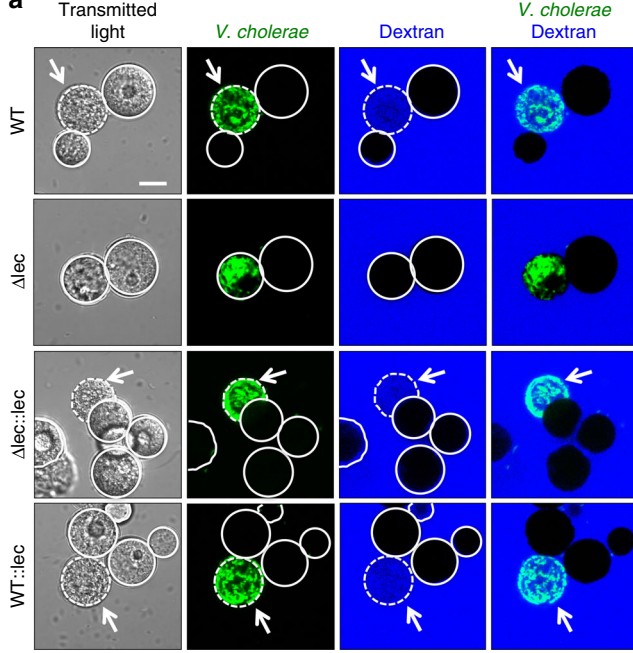

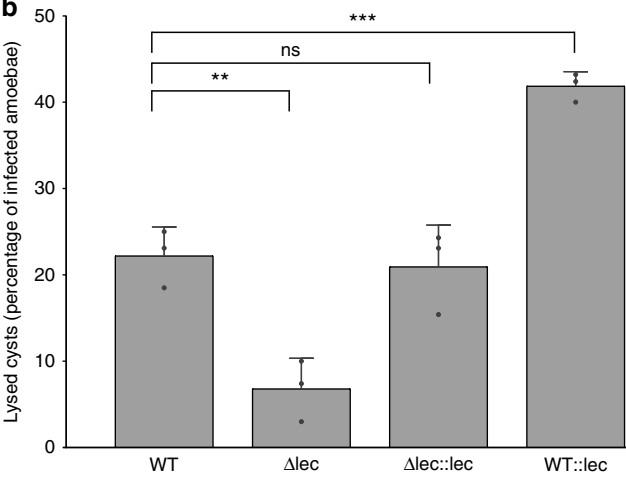

**Fig. 5** Cyst permeabilization by *V. cholerae* depends on its lecithinase. **a** Representative images and **b** quantification of lysed or non-lysed cysts that were infected by WT, lecithinase-minus (Δlec), lecithinase-minus but complemented (Δlec::lec), and *lec*-merodiploid *V. cholerae* strains (all GFP positive). The permeabilization phenotype was visualized through the infiltration of dextran from the surrounding medium in the white arrow-marked cysts, which did not occur for the Δlec strain. Scale bar: 10 μm. The graph in **b** represents average values from three independent biological replicates (±s.d.). Statistics are based on a one-way ANOVA with ***$p \leq$ 0.001; **$p \leq$ 0.01; ns, $p > 0.05$

interactions with non-human hosts. Such ancient host–pathogen interactions are, however, often considered as evolutionary precursors to modern interactions that occur between bacteria and their human hosts[60]. Moreover, aquatic predators are recognized for their contribution to pathogen emergence due to the selection pressure they exert on their bacterial prey[61]. Here, we examined the molecular mechanisms that *V. cholerae* uses to interact with the aquatic amoeba *A. castellanii* and to maintain a favorable replication niche within the amoebal osmoregulatory organelle. Based on the molecular checkpoints that were deciphered in the current study, we expanded our model of the pathogen's intra-amoebal lifecycle (Fig. 7). Specifically, we showed the importance of several

extracellular enzymes in this host–pathogen interaction. The production of the HapA protease, which cleaves the pore-forming hemolysin toxin, proved essential for avoiding premature intoxication of the amoebal host (Fig. 7, label A). In contrast to intracellular pathogens, such as *Listeria monocytogenes* or *Shigella flexneri*, that use PFTs to escape from acidic vacuolar compartments to reach the host cell's cytosol, the situation described herein is very different. In this case, the host–pathogen interaction relies on *V. cholerae* residing in a non-digestive vacuole in which it can readily replicate[25]. This non-digestive CV is an essential osmoregulatory organelle of the amoeba meaning that HlyA-mediated rupture of the vacuolar membrane would, therefore, release *V. cholerae* into the cytosol, though, at the expense of rapid host cell death. We speculated, therefore, that this HapA-mediated disintegration of HlyA might have evolved to allow *V. cholerae* to maximize its growth output within this intra-amoebal replication niche by avoiding the premature death of its host. Notably, the HapA protease is considered a minor virulence factor that contributes to disease outcomes in animal models of cholera[13,62], along with HlyA itself, as described above. In this study, we did not observe any obvious phenotype for *V. cholerae* strains lacking HlyA with respect to their ability to first colonize and then escape the amoebal CV. However, at this point we cannot exclude the contribution of HlyA as well as other, potentially redundant, enzymes to the other discussed phenotype in which the *V. cholerae* pathogen escapes from the *A. castellanii* phagosomal pathway[25]. Alternatively, HlyA might have a role in another context in the pathogen's environmental lifestyle. It was shown that HlyA causes developmental delays and lethality in the nematode *Caenorhabditis elegans* through intestinal cytopathic changes[63], and similar effect might occur with other hosts in the pathogen's aquatic reservoirs. Notably, both enzymes (HapA and HlyA) are widely distributed among pathogenic *Vibrios*[14], which reinforces the notion that they have important roles in environmental settings.

While the production of the HapA protease protects the CV from premature lysis, such lysis still occurs when the amoeba has undergone encystation and the CV is fully occupied by the bacteria. We have previously shown that the rupture of the vacuolar membrane is dependent on the production of the *Vibrio* extracellular polysaccharide (VPS; Fig. 7, label B), and we hypothesized that this might be due to a VPS-dependent agglutination effect[25]. Consistent with this idea is a recent study on the importance of the osmotic pressure generated by the hydrogel-like VPS matrix of *V. cholerae*[64]. These authors suggested that the capacity of *V. cholerae* to escape the amoebal CV, which we had previously reported[25], could indeed be related to the osmotic-pressure response of the VPS biofilm matrix[64]. Alternatively, the accumulation of the VPS might hinder the CV from contracting and therefore from expelling the accumulated water. Given the continued influx of water from the cytosol such a malfunctioning would likewise lead to the lysis of the vacuole, as observed (ref.[25] and this study) and consistent with a previous report in which the authors blocked the vacuole's contractility using antibodies against myosin-I[30].

After *V. cholerae* escaped from the vacuole and upon further growth within the cytosol of the amoebal cyst, another extracellular enzyme, the lecithinase/phospholipase, was required to ultimately kill the host and return the bacteria to the environment (Fig. 7, label E). This lecithinase is encoded in close proximity to the hemolysin gene, a region that was previously speculated to represent a pathogenicity island[20]. Unfortunately, apart from the absence of a phenotype for the *lec*-minus mutant in rabbit-ligated ileal loops[22], a model of cholera that primarily aims at judging cholera toxin-mediated fluid accumulation, this enzyme had not been extensively studied in vivo (for current 7th pandemic O1 El Tor strains). However, a recent study based on an activity-based protein profiling

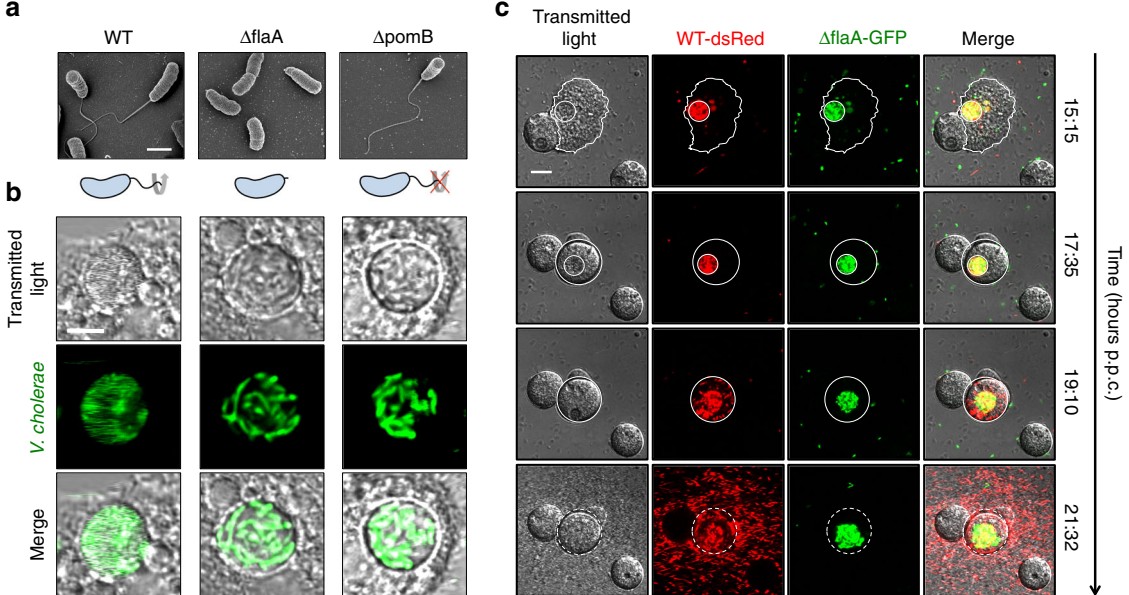

**Fig. 6** Efficient escape from the ruptured CV and the lysed cysts requires flagellum-based motility. **a** Scanning electron micrographs of WT *V. cholerae* and its *flaA*- and *pomB*-deficient derivatives, which lack the major flagellin and cannot rotate their flagellum, respectively (as schematized below the images). Scale bar: 1 μm. **b** Close-up view of confocal scanning images of CVs (transmitted light) that were colonized by each one of the three GFP-tagged strains shown in **a**. The scanning speed was lowered for these experiments to visualize the fast movement of WT *V. cholerae* inside the CV. See also Supplementary Movies 2 to 4 for dynamics of intra-vacuolar WT, ΔflaA, and ΔpomB bacteria. Scale bar: 5 μm. **c** Escaping from the lysed amoebal host and spreading requires flagellum-based motility. *A. castellanii* was infected with a 1:1 mixture of WT (dsRed-tagged) and non-motile *flaA*-minus (GFP-tagged) bacteria. Time-lapse microscopic imaging of a co-infected amoeba was started at 15 h p.p.c and followed for more than six hours (time is indicated on the right). Still images derived from the recorded movie (Supplementary Movie 5) are depicted and show (from top to bottom) the colonized trophozoite before and after encystation, after rupture of the CV, and after cyst lysis. Scale bar: 10 μm

method showed that the lecithinase/phospholipase is active in cecal fluids of infected infant rabbits[65]. Further studies are therefore required to conclusively determine whether this enzyme should also be considered as a minor virulence factor for animals, as we demonstrated here for amoebae.

Lastly, we showed that bacterial motility is required to efficiently escape from the lysed CV and the succumbed host (Fig. 7, labels C and D), which, ultimately, allows the bacteria to return to the aquatic environment. One could speculate that the lysed cyst might generate a nutrient gradient that attracts novel predators, a disadvantageous scenario for any prey. Indeed, we frequently observed that dead cysts were readily ingested by feeding trophozoites, which could counter-select for non-motile mutants that cannot escape from the cellular debris. Likewise, motility has also been demonstrated to be of importance for virulence in animal models. Indeed, non-motile mutants exerted less fluid accumulation in rabbit ileal loops[62] and were severely attenuated for the colonization of infant mouse small intestines[4]. These correlations between the contribution to virulence in animal models and in the environmental host–pathogen interaction described herein therefore support the "coincidental evolution hypothesis". This hypothesis suggests that "virulence factors result from adaptation to other ecological niches" and, in particular, from "selective pressure exerted by protozoan predator"[66].

## Methods

**Bacterial strains and growth conditions**. All bacterial strains used in this study are listed in Supplementary Table 1. *V. cholerae* strains are derivatives of the 7th pandemic O1 El Tor strain A1552 if not indicated otherwise. Bacteria were grown on LB plates (1.5% agar) or under shaking conditions in liquid LB medium. Antibiotics were supplemented when needed for genetic engineering and selection at the following concentrations: 100 μg/ml of ampicillin; 50 μg/ml of gentamicin, and 75 μg/ml of kanamycin. Amoebal infection media were free of antibiotics.

For counter-selection of site-directly modified phenylalanyl-tRNA synthetase (*pheS**)-carrying strains (Trans2 method; see below) 4-chloro-phenylalanine (cPhe;

C6506, Sigma-Aldrich, Buchs, Switzerland) was added to the medium before autoclaving. Optimization experiments showed that a concentration of 20 mM of cPhe was best for *V. cholerae* counter-selection.

**Genetic engineering of bacterial strains**. All oligonucleotides used for genetic engineering are listed in Supplementary Table 3. Genetic engineering of strains to delete/insert genes or to add gene fusions onto the *V. cholerae* chromosomes was performed using the previously described TransFLP method[67–70]. An alternative protocol, based on two rounds of transformation, was established in this study (named Trans2) to generate complemented strains or overexpression constructs. This method was adapted from Gurung et al.[71] to work in *V. cholerae* and is based on the counter-selectability of bacteria carrying a mutated version of the α subunit of phenylalanyl-tRNA synthetase in cPhe-containing medium. The gene encoding the phenylalanyl-tRNA synthetase of *V. cholerae* (α subunit; *pheS* or VC1219) was integrated into plasmid pBR-FRT-Kan-FRT2[51] to generate the plasmid pBR-FKpheSF (carrying the aminoglycoside 3′-phosphotransferase-encoding gene [*aph* (3′)] and *pheS* flanked by FRT sites). This and subsequent plasmids served as templates for inverse PCR to generate the site-directly mutated *pheS* in the pBR-FKpheSF[A294G] and pBR-FKpheSF[A294G/T251A] plasmids (renamed for simplicity as pFRT-aph-pheS*). This double-mutant template was generated based on a study in *E. coli* that showed improved counter-selectability of this version of pheS (*pheS**) on cPhe-containing agar plates[72]. In contrast to a previous study[71], the pheS* allele used in this study (pheS[A294G/T251A]) was placed downstream of the constitutively expressed *aph*(3′) to ensure expression independent of the subsequent insertion site on the chromosome. The *aph*(3′)-*pheS** construct was PCR amplified using the pFRT-aph-pheS* plasmid as a template and fused by overlapping PCR with fragments carrying parts of the 5′- and 3′-region of the *lacZ* gene. The resulting PCR fragment (*lacZ*′-*aph*(3′)-*pheS**-′*lacZ*) was used as a transforming material for chitin-induced naturally competent *V. cholerae* cells[68]. Transformants were selected on kanamycin-containing LB agar plates. The growth impairment of the transformants due to the presence of *pheS** was verified on cPhe-containing agar plates followed by verification through Sanger sequencing. These strains underwent a second round of transformation on chitin (therefore the name Trans2). The second-round transforming material was devoid of any resistance gene and carried the desired genetic construct (generated by PCR amplification) flanked by the 5′- and 3′-parts of the *lacZ* gene. Screening for the integration of this construct was done using cPhe-containing agar plates. Transformants that grew were tested by colony PCR for the replacement of the *aph*(3′)-*pheS** genes and, ultimately, confirmed for marker- and scar-less integration using Sanger sequencing.

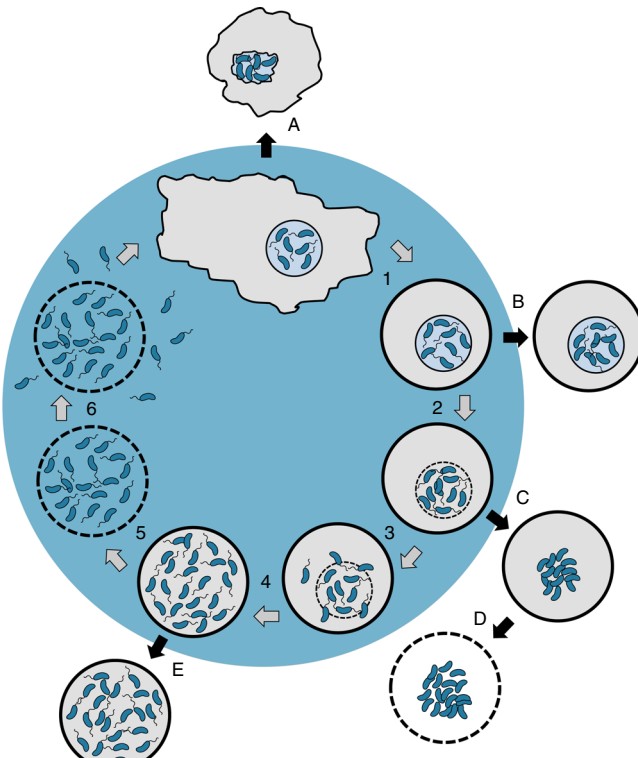

**Fig. 7** Model depicting the herein-described molecular checkpoints of *V. cholerae*'s amoebal infection cycle. After phagocytosis, *V. cholerae* cells colonize the CV of *A. castellanii* trophozoites. The bacteria remain within this replication niche upon amoebal encystation (1). After intra-vacuolar growth, rupture of the vacuole occurs in a *Vibrio* polysaccharide-dependent manner (2), which releases motile *V. cholerae* into the cyst's cytosol (3) where the bacteria proliferate further (4). Lecithinase-mediated membrane permeabilization (5) eventually leads to the lysis of the cyst and the quick release of the motile bacteria (6), which can undergo another round of infection. Defined bacterial mutants are blocked at different checkpoints and, therefore, impaired in the progression of the infection cycle: A, protease-deficient and therefore hemolysis-overactive strain (ΔhapA); B, *Vibrio* polysaccharide (VPS)-deficient strain (ΔvpsA[25]); C-D, non-motile strains (ΔflaA and ΔpomB); and E, lecithinase-minus strain (Δlec)

Variants of the mini-Tn7 transposon carrying constitutively expressed *gfp*, an optimized version of *dsRed* (*dsRed.T3[DNT]*[50,73]), or mCherry were stably integrated into the *V. cholerae* chromosome through triparental mating[74].

**Determination of enzymatic activities.** Semi-quantitative assays in specific media were used to determine enzymatic activities. To perform these assays, 5 μl of the respective overnight bacterial cultures were spotted onto the following three types of agar plates. (i) Milk agar was used to determine protease activity due to the hydrolysis of casein. The modified recipe used in this study contained 0.5% tryptone, 0.25% yeast extract, 0.1% dextrose, 1% skim milk powder, and 1.25% agar. The bacteria were incubated on these plates for 24 h at 30 °C. (ii) Trypticase Soy Agar II with 5% sheep blood (BD, Heidelberg, Germany) was used to test for hemolysis (24 h at 30 °C). (iii) Egg yolk agar plates (Hardy Diagnostic, Santa Maria, CA, USA) were used to evaluate lecithinase activity (72 h at 30 °C). Lecithinase activity was quantified after taking pictures with a tabletop scanner and analyzing these images using ImageJ[75]. Values are given according to the formula (area of bacteria + area of precipitate) − (area of the bacteria) = area of precipitate [cm²].

**Gene expression analysis based on qRT-PCR.** The relative gene expression compared to the gene *gyrA* was determined using quantitative reverse transcription PCR (qRT-PCR)-based transcript scoring in *V. cholerae*[76]. Data are based on three biologically independent experiments (±standard deviation [s.d.]). Primers used for qPCR are indicated in Supplementary Table 3.

**Amoebal infections.** The amoeba *Acanthamoeba castellanii* (ATCC strain 30010) served as the host in all of the infection experiments. Supplemented peptone-yeast-glucose (PYG) medium (ATCC medium 712) was used to propagate uninfected amoebae and half-concentrated defined artificial seawater (0.5× DASW; buffered with 50 mM HEPES)[77] was used as the infection medium.

For infection experiments, amoebae were diluted with fresh PYG at a concentration of $1 \times 10^5$ amoebae/ml and seeded into a μ-Dish (low wall 35 mm ibidiTreat devices; 80136-IBI, Vitaris, Baar, Switzerland). After three hours of static incubation at 24 °C, adherent amoebae were washed three times with the infection medium. Exponentially growing bacteria were likewise extensively washed in infection medium and added to the amoebae at a multiplicity of infection (MOI) of 1000. Co-cultures were incubated statically at 24 °C for the indicated time post-primary contact (p.p.c.)[25].

**Confocal laser scanning microscopy-based techniques.** Confocal lasers scanning microscope (CLSM) imaging was done using a Zeiss LSM 700 inverted microscope (Zeiss, Feldbach, Switzerland). To label the endosomal pathway of the amoebae, Alexa Fluor 647-labeled dextran (MW 10,000; D22914, Thermo Fisher Scientific, Waltham, Massachusetts, USA) was added to the infection medium at a final concentration of 100 μg/ml. The fluorescent stain calcofluor white was supplemented wherever indicated at a final concentration of 0.2%, as this stain is known to bind to cellulose (18909-100ML-F, Sigma-Aldrich, Buchs, Switzerland). Fluorescent signals were quantified using the open-source imaging software ImageJ[75] (NIH, Bethesda, MD, USA). Bacterial motility inside the amoebae (see Supplementary Movies 2, 3, and 4) was assessed using short frame intervals and/or a reduced scanning speed.

**Scanning electron microscopy.** For scanning electron microscopy (SEM) imaging, a 10 mm in diameter silicon wafer was coated with 100 μg/ml of poly-D-lysine hydrobromide (P6407, Sigma-Aldrich, Buchs, Switzerland) for 2 h at room temperature, then washed three times with bi-distillated water. Exponentially grown bacteria were washed with PBS and allowed to settle for 20 min at room temperature onto the silicon wafer. The supernatant was removed, and a 0.1 M phosphate buffer solution of 1.25% glutaraldehyde and 1% tannic acid was gently added. After 1 h, the fixative was removed and replaced by cacodylate buffer. This was transferred to a 1% solution of osmium tetroxide in cacodylate buffer for 30 min, washed in water, and then dehydrated through a series of increasing concentrations of alcohol. The sample was finally dried at the critical point of carbon dioxide and coated with a 2-nm-thick layer of osmium metal using an osmium plasma coater. Images were collected in an SEM (Merlin, Zeiss NTS) at 1.8 kV electron beam tension.

**Correlative light and electron microscopy.** For the correlative light and electron microscopy (CLEM), 18 mm glass coverslips were used. These were coated with a thin (10 nm) layer of carbon onto which an alphanumeric grid had been outlined using a thicker, 20 nm, carbon layer. The coverslips were incubated with poly-D-lysine hydrobromide (P6407, Sigma-Aldrich, Buchs, Switzerland) at a concentration of 100 μg/ml for 30 min at room temperature and then washed three times with PBS. At 20 h p.p.c., colonized amoebae were detached from the tissue culture flask using a cell scraper, plated onto coated coverslips, and allowed to adhere for 30 min at room temperature. The supernatant was removed, and a buffered aldehyde solution of 1% glutaraldehyde and 2% paraformaldehyde was gently added. After one hour at room temperature, these were washed in cacodylate buffer. The corresponding confocal images were acquired. The samples were then post-fixed and stained with 1% osmium tetroxide followed by 1% uranyl acetate. After dehydration through increasing concentrations of alcohol, the coverslips were embedded in EPON resin and polymerized at 65 °C for 48 h. Cells of interest were located using the alphanumeric grid that remained on the resin surface after the grid had been removed. Blocks were trimmed around the cell and between 200 and 300 serial thin sections, at a thickness of 50 nm, were cut from the entire structure using a diamond knife and ultra-microtome. Sections were collected onto copper slot grids, carrying a formvar support film, and then further stained with uranyl acetate and lead citrate. Serial images of the cell were taken with a transmission electron microscopy (TEM; Tecnai Spirit, Fischer Scientific) at a voltage of 80 kV.

**Estimation of number of vacuolar bacteria.** The number of bacteria in the vacuole was estimated using an unbiased stereological approach[31] by counting bacteria in five probes across a series of 55 TEM sections cut through the middle of a single vacuole. Each section was 50 nm thick and each probe consisted of a reference section and a look up section that were five sections apart. The volume of each probe was 8.3 cubic micrometers. From these counts we calculated a mean density of 1.35 ± 0.42 bacteria per cubic micron. Given the approximate total volume of the CV of 143.8 μm³, based on the assumption of a spherical shape, the total number of *V. cholerae* cells was estimated to be 194 (±60). While this experimental approach was elaborate and, therefore, not applicable to large numbers of individual amoeba, an average number of 150–200 *V. cholerae* cells per organelle is also theoretically supported given an average volume of ~1 μm³ (www.BioNumber.org) and 150 μm³ (based on a maximum diameter of up to 6.6 μm[78]) for bacteria and the CV of *A. castellanii*, respectively.

To generate a 3D model of the colonized CV, serial TEM images were first aligned manually using image software (Photoshop, Adobe). Each structure was then segmented using TrakEM2[79] operating in the FIJI software[80] (www.fiji.sc).

The 3D models were then exported into the Blender software (www.blender.org) for smoothing and rendering into a final image or movie.

**Epifluorescence microscopy**. To visualize T6SS production and assembly in *V. cholerae*, epifluorescence microscopy images were acquired using a Zeiss Axio Imager M2 microscope using standard settings[50,76]. Images in Fig. 4c were taken with the same exposure times for the green (GFP) channel.

**Interbacterial predation assay**. T6SS-mediated killing of *E. coli* prey by diverse *V. cholerae* strains was performed by co-incubation on solid LB agar surfaces. Recovered prey was enumerated on selective medium[50,51].

**Statistical analysis**. One-way ANOVA was performed using GraphPad Prism version 7 for Mac (GraphPad Software, San Diego, CA, USA) followed by Tukey's post hoc test for multiple comparisons. Statistical analyses were performed on three biologically independent experiments. $*p \leq 0.05$; $**p \leq 0.01$; $***p \leq 0.001$; $****p \leq 0.0001$; ns (not significant), $p > 0.05$. The amoebal population counted manually was each time 6000 (derived from three biologically independent experiments each with $n = 2000$).

**Data availability**. All data supporting the findings of this study are available from the corresponding author upon request.

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

## Acknowledgements

We thank members of the Blokesch group at EPFL and the Geneva/Lausanne amoeba club for fruitful discussions. We also acknowledge J. Mekalanos (Harvard) for strain C6706 (original stock) and former members of the Blokesch lab for provision of genetically engineered *V. cholerae* strains. This work was supported by EPFL intramural funding, a Starting Grant from the European Research Council (ERC; 309064-VIR4-ENV), and a Consolidator Grant from the European Research Council (ERC; 724630-CholeraIndex) to MB. M.B. is a Howard Hughes Medical Institute (HHMI) International Research Scholar (grant 55008726).

## Author contributions

C.V.d.H. and M.B. designed the study; C.V.d.H., A.S.V., N.C.D.D., G.K., and M.B. planned the experiments; C.V.d.H., A.S.V., N.C.D.D., S.S., C.S., T.S., S.C., C.M., and M.B. performed experiments; C.V.d.H., A.S.V., N.C.D.D., G.K., and M.B. analyzed the data; C.V.d.H. and M.B. wrote the manuscript; M.B. revised the manuscript. All authors approved the final version of the manuscript.

## Additional information

**Competing interests:** The authors declare no competing interests.

