## [Peer Review File · Nature Communications]

Reviewers' comments:

Reviewer #1 (Remarks to the Author):

SUMMARY

Van der Henst et al. show that HapA protease regulates the pore-forming hemolysin HlyA which is responsible for amoebal over-intoxication. Furthermore, the authors show the function of the lecithinase and motility in escaping the amoeba. This manuscript uses an elegant approach, combining microscopy and bacterial genetics. The presented data supports all their claims and presents an exciting way to use CLEM to study the intracellular component of a bacteria's lifecycle. The paper is very well written.

MAJOR COMMENTS

1. It would be helpful to see translational fusions of the promoter of the 5 genes (hapA, hlyA, vspA, lecA and flaA) with a destabilized fluorescent protein to get a better understanding of the spatial-temporal relationship of these proteins inside the host. Are the proteins only made at their appropriate time (as shown in figure 6), or are they all constantly expressed and other factors are responsible for the progression of the infection? This would go a long way to supporting the model.
2. Is this a general property that applies to other cholera strains (both pandemic and non-pandemic) and different species of amoeba.
3. What is the role of the T6SS considering the expertise of the authors? It would be sufficient to mention T6SS and RTX in the section about accessory toxins

MINOR COMMENTS

1. Please introduce the cholera strain you are using in the main text
2. Page 5 describes the experimental setup in quite some detail and could be moved to the Materials & Methods section if space is needed.
3. Line 131: please add statistical error (194 +/- X)
4. Lines 59-69. What is the role of hlyA in pandemic strains as pandemic strains are used in this paper?
5. Is lecA also degraded by HapA? How is lecA controlled to be active at the right moment?
6. The authors should acknowledge the limitations of their "educated guess" approach

Reviewer #2 (Remarks to the Author):

In the manuscript entitled "Molecular insights into *Vibrio cholerae*'s intra-amoebal host-pathogen interactions," the authors describe the role of the *V. cholerae* hemagglutinin protease (HAP), hemolysin A, lecithinase (LecA), and the flagellum in persistence and escape of the contractile vacuole (CV). HAP attenuates virulence by degrading HlyA, which destroys the contractile vacuole, leading to amoebal death. *V. cholerae* colonizes the CV and remains there during encystation, escapes the CV by synthesis of the VPS exopolysaccharide, and then degrades the cyst membrane by means of the lecithinase, lecA. After escape, the flagellum allows bacterial dispersal. These are interesting and original findings that provide a role in amoebal infection for many pathogen proteins that do not play an important role *V. cholerae* pathogenesis in the mammalian intestine. The experiments are well-described, carefully performed, and conclusive. However, the manuscript could benefit from additional background that would allow the reader to evaluate these findings critically. Specific comments follow.

1) Introduction: The reader would benefit from more detail regarding the amoeba and the behavior of its intracellular vacuoles in the absence of infection. These amoebae live in fresh water. My understanding is that the CV is used to maintain intracellular homeostasis in the face of an intracellular osmotic strength that is higher than that of their fresh water surroundings. Here are some questions that were left unanswered by the introduction and might help the reader put the findings in context:

a) The authors should explain the role of the contractile vacuole in more detail and what is known about the intravacuolar ionic strength and osmolarity. Given its function, one might assume the intravacuolar osmolarity is quite low. Is this compatible with the salt requirements of *V. cholerae*, or must one presume an alteration of the intravacuolar environment by *V. cholerae*? How does the hyperosmolar environment under which these infections are performed affect the conditions within the CV?

b) The life cycle of the amoebae should be described. Is encystation a baseline process or is it the result of infection in the high salt medium used for infection. Does *V. cholerae* alter the kinetics of encystation?

c) The findings of the previous manuscript should be discussed more clearly along with the open questions that are these were addressed here.

d) The focus of the current introduction on the role of these minor virulence factors in mammalian hosts might be better saved for the discussion, where it can be compared with their role in the amoeba.

2) Even after reading both the published and current manuscripts, I am confused as to how *V. cholerae* accesses the CV. Is it creating a new pathway or co-opting an existing one?

3) Please comment on the proportion of amoeba that become infected with *V. cholerae*. The requirement for the CLEM technique suggests that this is a very low frequency event.

4) Amoeba infected with HapA- strains are noted to have aberrant contractile vacuoles and do not encyst. If I understand this correctly, the authors conclude that the amoeba do not encyst because they are not viable. Is a direct test of the viability of HapA-infected amoebae possible? For instance, are markers of apoptosis present or is the cell membrane permeability increased?

5) Line 127: Please define the term dissector pairs and give units if any for the value 1.35.

6) Line 258: This observation suggests that HapA results in complete degradation of GFP rather than just cleavage of HlyA with preservation of fluorescence. Is this the case? Western analysis would confirm complete degradation of the fusion protein.

7) Line 260: GFP alleles are notorious for aggregation, which results in mislocalization within cells. Because there is no evidence that this distribution of GFP is not an artifact, the authors should be more circumspect.

8) Line 330: Transcription of *hapA* is activated at high cell density and that of *hlyA* is repressed. At the same time, HapA degrades HlyA. This suggests that HlyA might play a role early in infection before high cell density is reached. Here, however, HlyA does not appear to be required for infection. How does the regulation of these two proteins fit in with the observations presented here and what might the role of HlyA be prior to activation of HapA?

9) Line 348: The authors have previously noted that escape from the CV is dependent on VPS. While this might be the result of osmotic pressure generated by VPS, more must be known about the

physicochemical characteristics of the CV and the cytoplasm to evaluate this possibility. Furthermore, a defect in exopolysaccharide production has been shown to impact the transcription of many genes. Therefore, another possibility is that deletion of the vps genes decreases expression of factors required for CV escape.

10) Where cell morphology was assessed, please comment in the methods section on whether the 6,000 cells counted were assessed manually or using a morphological analysis program.

Rebuttal letter

Molecular insights into *Vibrio cholerae*'s intra-amoebal host-pathogen interactions

First of all, we would like to thank the editor and the two referees for the time they spent on reviewing our manuscript. We very much appreciate the comments and suggestions. Based on the provided feedback, we were able to further improve the manuscript. Below, we provide point-to-point answers explaining how we addressed the comments. We truly believe that we have addressed all technically feasible experiments that are within the scope of the current study. Thank you for taking the time to go through this rebuttal letter and the revised manuscript.

>> Reviewer #1

2) SUMMARY

Van der Henst et al. show that HapA protease regulates the pore-forming hemolysin HlyA which is responsible for amoebal over-intoxication. Furthermore, the authors show the function of the lecithinase and motility in escaping the amoeba. This manuscript uses an elegant approach, combining microscopy and bacterial genetics. The presented data supports all their claims and presents an exciting way to use CLEM to study the intracellular component of a bacteria's lifecycle. The paper is very well written.

>> Authors' comment: We very much appreciate the kind words of the reviewer. Concerning the revised manuscript, we hope that our new experiments are satisfactory to the reviewer, even though we realized that some of the envisioned experiments were technically impossible.

MAJOR COMMENTS 3)

1. It would be helpful to see translational fusions of the promoter of the 5 genes (hapA, hlyA, vspA, lecA and flaA) with a destabilized fluorescent protein to get a better understanding of the spatial-temporal relationship of these proteins inside the host. Are the proteins only made at their appropriate time (as shown in figure 6), or are they all constantly expressed and other factors are responsible for the progression of the infection? This would go a long way to supporting the model.

>> Authors' comment: We thank the reviewer for this interesting comment. First, we would like to mention that we do not claim that the expression of these genes is adapted to the specific niche (e.g., the contractile vacuole). Instead, we believe that the effect of the proteins might be adapted, which is what ultimately counts and what we addressed in this study. This can be based on expression, protein stability, but also accumulation in a closed environment such as the contractile vacuole or the cytosol of the cyst. For example, it has been known for more than 15 years (shown by the groups of Bassler, Yildiz, Mekalanos) that the *Vibrio* polysaccharide (*vps*) gene cluster is regulated by quorum sensing (repressed at high cell density). As such, the bacteria would produce *Vibrio* polysaccharide until a certain threshold of bacteria is reached and then stop making more matrix material. However, due to the closed environment inside the amoebae, the VPS cannot be washed away and therefore accumulates within the vacuole. The HapA protease is likewise known to be regulated by quorum sensing but in an opposite manner, meaning it is solely produced at high cell density.

It should be noted that the destabilized version of GFP, as suggested by the reviewer, has been widely used before in the group in which the senior author performed her postdoc (Schoolnik lab at Stanford) and in her own group and we therefore knew that the signal for destabilized GFP is usually very weak. Consequently, this approach is only reliable for strongly expressed promoters (e.g., such as the *tcpA* promoter in *V. cholerae*'s virulence program (Nielsen et al., 2010; PLoS Pathogen) or a strong ribosomal promoter (e.g., P1 promoter of the *rrnB* ribosomal operon), which is often used to monitor growth (Nielsen et al., 2010, PLoS Pathogen; Van der Henst et al., 2016, ISME J).

To still address the reviewer's comment, we performed the suggested experiment. Precisely, we

generated five fusion constructs: a negative control (no promoter) and four fusions between the promoters of *hapA*, *hlyA*, *vpsA*, and *lecA* and a fluorescent protein reporter gene. We did not include *flaA*, as the flagellar activity can be directly observed through motility of the bacteria through time-lapse microscopy and we do not expect a direct correlation *between* *flaA* expression (or lack of it) and motility. As reporter gene, we used destabilized GFP [GFP(ASV)], as suggested by the reviewer. Next, we inserted these fusion constructs into the chromosome, as plasmid-based systems are mostly non-reliable in *V. cholerae* (and might not be maintained when the bacteria grow inside the amoebae).

We tested all fusion by checking expression of GFP(ASV) under rich growth conditions at low cell density and at high cell density. All strains also produced mCherry constitutively, as this would allow us to localize the bacteria inside the amoeba! host. Below, you can see the results of the six different strains that we tested (including a *gfp*-minus control). Each strain is shown in the phase contrast channel (top), the green channel (GFP(ASV) signal) and the red channel (constitutive mCherry signal).

As the reviewer (and editor) can hopefully appreciate from the images below (panel A), the expression of most of the tested promoters is too low to allow us to draw any conclusions with respect to the expression level. Notably, the epifluorescent images in the green channel were taken with 1000msec exposure time, which further confirms the low level of green fluorescence. Such long exposure times do not allow proper imaging using the laser scanning confocal microscope, due to the movement of the bacteria inside the amoeba. We nonetheless tried for the *vpsA* fusion, which, *in vitro*, had shown some fluorescent signal. However, as expected, the green signal was extremely low throughout the experiment (panel B). We therefore realized that the suggested experiments are technically impossible. Apart from this fact, and as mentioned above, we do not claim that the expression of these genes is triggered at a certain time inside the amoebae but that this might follow the normal expression patterns, including quorum sensing-mediated induction or repression.

A

Low cell density

Low cell density

High cell density

WT
mcherry

[-]-gfp(ASV)
mcherry

[vpsA]-gfp(ASV)
mcherry

High cell density

[hlyA]-gfp(ASV)
mcherry

[lec]-gfp(ASV)
mcherry

[hapA]-gfp(ASV)
mcherry

Figure: Reporter fusion imaged after *in vitro* growth or inside infected amoebae. (A) Bacteria were grown in rich medium to a low cell density (upper part) or high cell density (lower part) and then imaged by epifluorescent microscopy. The images shown represent the phase contrast channel (top), the green channel (middle), or the red channel (bottom). The strains tested were: WT mCherry: strain with constitutive *mCherry* expression and without a *gfp(ASV)* gene (negative control for background fluorescence); [-]-*gfp(ASV)* mCherry: strain with constitutive *mCherry* expression and promoterless *gfp(ASV)*; [*vpsA*]-*gfp(ASV)* mCherry: strain with constitutive *mCherry* expression and *gfp(ASV)* preceded by *vpsA* promoter; [*hlyA*]-*gfp(ASV)* mCherry: strain with constitutive *mCherry* expression and *gfp(ASV)* preceded by *hlyA* promoter; [*lee*]-*gfp(ASV)* mCherry: strain with constitutive *mCherry* expression and *gfp(ASV)* preceded by *lee* promoter; [*hapA*]-*gfp(ASV)* mCherry: strain with constitutive *mCherry* expression and *gfp(ASV)* preceded by *hapA* promoter. (B) Confocal laser scanning microscopy of the strain [*vpsA*]-*gfp(ASV)* mCherry at 20 hours p.p.c. Images shown represent the red channel (top left), green channel (top right), the transmitted light channel (lower left) and a merged image of all channels (lower right). Amoebae infected by *V cholerae* inside their contractive vacuole are shown.

4)

2. Is this a general property that applies to other cholera strains (both pandemic and non-pandemic) and different species of amoeba.

>> **Authors' comment:** Again, we thank the reviewer for this interesting question. We were especially keen on answering the first part of the comment, namely how general the observed properties are in other pandemic *V cholerae* strains and also in environmental isolates. We therefore compared three different pandemic strains: A1552 (an O1 El Tor isolate from a Peruvian outbreak in the 1990ies), strain C6706 (another O1 El Tor isolate from a Peruvian outbreak in the 1990ies; we received the original stock from John Mekalanos, as one of the currently circulating strains has a lab-derived quorum sensing defect, which masks very important phenotypes [see Stutzmann and Blokesch, 2016]), and the reference genome strain N16961 (an O1 El Tor isolate from Bangladesh in the 1970ies). Please note that we used a quorum-sensing proficient variant of the latter strain by repairing the single bp frameshift mutation. We tested the WT parental strains but also their *hapA*- and *hapA/hlyA-minus* derivatives. As you can see in the new supplementary figures, all three pandemic strains behaved similar with respect to HlyA-mediated amoebal intoxication in the absence of the HapA protease (Fig. S5) and the impaired cyst lysis in the absence of the lecithinase (Fig. S6).

With respect to non-pandemic strains, we chose an environmental isolate from the California coast (strain Sa5Y), as this strain is amenable to genetic engineering through natural transformation. This strain showed constitutive expression of *hlyA*, which might reflect adaptation to its natural environment. However, consistent with what we showed for the pandemic strains, the strain also leads to aberrant morphology of the amoebae, a phenotype that is fully dependent of the hemolysin protein. These data are also included in Fig. S5. Moreover, this environmental strain also required the lecithinase to efficiently escape from the amoebal cysts.

Concerning the second part of the question: we are also working with the social amoeba *Dictyostelium discoideum*. But we expect that the osmoregulation is in general very different between the two amoebal species *A. castellanii* and *D. discoideum*, as the latter lives in soil. Thus, it is unlikely that *V cholerae* would have adapted to *D. discoideum*, as the two organisms probably do not encounter each other in nature. Unfortunately, we cannot conclusively study CV colonization of *D. discoideum*, as we see a strong toxicity of *V cholerae* towards *D. discoideum* in liquid co-cultures (*unpublished and confidential information*).

5)

3. What is the role of the T6SS considering the expertise of the authors? It would be sufficient to mention T6SS and RTX in the section about accessory toxins.

>> **Authors' comment:** This is indeed a very interesting question and of prime importance, especially as both the T6SS and the RTX have pore-formation activity, as does HlyA. Concerning RTX: We know that RTX is produced in pandemic *V cholerae* and we therefore tested the *rt.x-minus* mutant in the amoebal infection assay. We did not observe any change in amoebal morphologies or a changed dynamics of *V cholerae*'s intra-amoebal lifecycle (new Fig. 4, panel A).

Concerning the T6SS: it turns out that this question often comes up after presentations. This isn't too surprising given the history of the type VI secretion system, which was discovered by Stefan Pukatzki then in John Mekalanos' lab because of its toxicity towards the soil amoeba *D. discoideum* (Pukatzki et al., 2006, PNAS). However, the effect of the T6SS on aquatic amoebae, which co-inhabit the same environmental reservoirs as *V cholerae*, was, to our knowledge, never tested. We initially did not include the data, as the T6SS seems to have no effect in pandemic strains in the here-described interaction model with *A. castellanii*. However, as this is a recurrent question and probably of interest to many researchers, we now re-tested this in-depth and included the data as new Fig. 4.

Notably, the initial study on the T6SS discovery (Pukatzki et al., 2006, PNAS) and several of the follow-up studies were based on the non-pandemic strain V52, a toxigenic 037 serogroup isolate from the 1960ies. This strain has a constitutively activate T6SS. This is in contrast to the pandemic 01 El Tors strains, which are T6SS silent under standard laboratory conditions and require specific environmental cues such as chitin (Borgeaud et al., 2015, Science) or low c-di-GMP concentration (Metzger et al., 2016, Cell Rep) to significantly induce the system. Thus, to better test whether the T6SS system might have an effect on *A. castellanii* under conditions under which the system would be on, we tested a constitutively active T6SS strain, ATCC25872. This strain is a toxigenic 037 serogroup strain isolated in the 1960ies and therefore closely related to strain V52. However, in contrast to strain V52, which has a premature stop codon in the gene encoding the major regulator of quorum sensing (HapR; Chun et al., 2009, PNAS), strain ATCC25872 is quorum sensing proficient and therefore positive for HapA production and amenable to genetic engineering by natural transformation (Blokesh and Schoolnik, 2007). As can be seen in Fig. 4, this strain shows constitutive T6SS activity and interbacterial killing of non-kin bacterial cells. However, the T6SS activity has no consequence on the amoebal host. Indeed, the amoeba showed normal morphologies and the dynamics of the *V cholerae* lifecycle was comparable to the pandemic control strain (A1552; Fig. 4F). We therefore conclude that the T6SS has no impact on *A. castellanii* under the tested conditions.

MINOR COMMENTS

6)

1. Please introduce the cholera strain you are using in the main text.

>> **Authors' comment:** We now provide detailed information on the *V cholerae* strain we are primarily using in our group in the main text, as suggested by the reviewer and consistent with the comparative data for the 3 different pandemic isolates (A1552, C6706, and N16961). Please note that the pandemic strain we are working with, A1552, is commonly used in many laboratories. A1552 was isolated in the US from a traveller returning from South American in 1992 and it belongs to the 01 El Tor Inaba group. Based on older comparative genomics using microarrays (Blokesch and Schoolnik 2007) and RNAseq data (Borgeaud et al., 2015) we know that it is very similar to other 7th pandemic strain, consistent with a recent report by Domman et al., 2017 (Science) on several Peruvian cholera isolates.

7)

2. Page 5 describes the experimental setup in quite some detail and could be moved to the Materials & Methods section if space is needed.

>> **Authors' comment:** We thank the reviewer for this comment and considered moving some details to the methods section. However, as this method was specifically developed for the current study and given that the description is not too extensive, we prefer to keep the information in the result section. This doesn't change the overall length of the complete manuscript (as the methods section is also part of the main manuscript).

8)

3. Line 131: please add statistical error (194 +/- X).

>> **Authors' comment:** The statistical error of the overall number has been added to the text (194±60.5).

9)

4. Lines 59-69. What is the role of *ohlyA* in pandemic strains as pandemic strains are used in this paper?

>> **Authors' comment:** We thank the reviewer for this comment. Based on the comment, we realized that parts of this paragraph weren't clear in the initial manuscript. In fact, most of the paragraph dealt with pandemic strains and solely the study by Ichinose *et al.*, 1987 was related to non-pandemic strains. We have clarified this in the revised manuscript by mentioning that the other studies were based on pandemic strains.

10)

5. Is *lecA* also degraded by HapA? How is *lecA* controlled to be active at the right moment?

>> **Authors' comment:** These are excellent questions. Unfortunately, we don't have answers to them at this stage. Concerning the first part of the question: based on the experimental data, it is unlikely that *Lee* is also degraded by the protease, as HapA is produced early on, as seen by the toxic effect of the mutant strain compared with the WT parental. The presence of HapA at this stage would therefore inactivate *Lee* early on. If this were the case, then a *lee*-minus strain wouldn't be expected to show the strong phenotype that we observe compared to the WT parental, as the protein would be degraded in the WT strain. It might be worth mentioning that we raised antibodies against *Lee*-derived peptides but, unfortunately, none of the antibodies recognized the protein.

Concerning the regulation: in fact, apart from activity measurements in the past, very little is known about this protein and about its regulation. However, we would like to point out that we do not claim that the *lee* gene is specifically induced shortly before the lysis of the cysts. In fact and consistent with point #3 above, it is more likely that the protein is produced by the bacteria early on and only reaches its target (e.g., the plasma membrane) after the contractile vacuole has lysed at the late step of infection. Together with the absence of a functional osmoregulatory organelle (the contractile vacuole) in the amoebae at this time, severe damage might be inflicted by the enzyme and result in the final death of the amoeba.

11)

6. The authors should acknowledge the limitations of their "educated guess" approach.

>> **Authors' comment:** We appreciate this comment and acknowledge the limitations of this approach in the revised manuscript.

>> **Reviewer #2**

12)

In the manuscript entitled "Molecular insights into *Vibrio cholerae*'s intra-amoebal host-pathogen interactions," the authors describe the role of the *V. cholerae* hemagglutinin protease (HAP), hemolysin A, lecithinase (LecA), and the flagellum in persistence and escape of the contractile vacuole (CV). HAP attenuates virulence by degrading HlyA, which destroys the contractile vacuole, leading to amoebal death. *V. cholerae* colonizes the CV and remains there during encystation, escapes the CV by synthesis of the VPS exopolysaccharide, and then degrades the cyst membrane by means of the lecithinase, LecA. After escape, the flagellum allows bacterial dispersal. These are interesting and original findings that provide a role in amoebal infection for many pathogen proteins that do not play an important role in *V. cholerae* pathogenesis in the mammalian intestine. The experiments are well-described, carefully performed, and conclusive. However, the manuscript could benefit from additional background that would allow the reader to evaluate these findings critically. Specific comments follow.

>> **Authors' comment:** We thank the reviewer for the kind words concerning the interesting and original findings of this study. And we very much appreciated the detailed feedback. Addressing the points raised by the reviewer has certainly improved the manuscript. Please find below our answers.

13)

1) Introduction: The reader would benefit from more detail regarding the amoeba and the behavior of its intracellular vacuoles in the absence of infection. These amoebae live in fresh water. My understanding is that the CV is used to maintain intracellular homeostasis in the face of an intracellular osmotic strength that is higher than that of their fresh water surroundings. Here are some questions that were left unanswered by the introduction and might help the reader put the findings in context:

a) The authors should explain the role of the contractile vacuole in more detail and what is known about the intravacuolar ionic strength and osmolarity. Given its function, one might assume the intravacuolar osmolarity is quite low. Is this compatible with the salt requirements of *V. cholerae*, or must one presume an alteration of the intravacuolar environment by *V. cholerae*? How does the hyperosmolar environment under which these infections are performed affect the conditions within the CV?

>> **Authors' comment:** We thank the reviewer for the interesting comments and questions.

Please note that *Acanthamoeba* are widely spread and not limited to fresh water conditions. Indeed, Khan (2002; *Encyclopedia of Life Sciences*) wrote that the genus *Acanthamoeba* "are ubiquitous in their distribution and can be found in freshwater, seawater, tap water, bottled mineral water, laboratory distilled water wash bottles, chlorinated swimming pools, air, soil and sewage."

Concerning the contractile vacuole: As written in Richard Allen's publication on "The contractile vacuole and its membrane dynamics" (Allen, 2000; *BioEssays* 22:1035-1042): "The contractile vacuole (CV) is an osmoregulatory organelle whose mechanisms of function are poorly understood". Thus, most of the questions raised by the reviewers cannot be answered at this point, as the contractile vacuole itself is poorly studied. This might be the case, as researchers have neither succeeded to genetically manipulate *A. castellanii* nor use many of the common inhibitors, which is a clear disadvantage of using this amoeba as a model system. However, due to its aquatic nature it is the most biologically relevant amoebae for our studies on *V. cholerae*, as both organisms have been detected in the same aquatic environments in cholera endemic areas (Shanan *et al.*, 2011).

To still address as much as possible the reviewer's questions, we now added a detailed description of what is known about the contractile vacuole to the introduction.

With respect to "Is this compatible with the salt requirements of *V cholerae*" => please note that *V cholerae* can grow in fresh water, as for example witnessed in Neusiedler lake in Austria (see Kirschner et al., 2008, AEM and 2011, Microb. Ecol.) and the Great African lakes in the Democratic republic of Congo (DRC) (Bompangue et al., 2009, PLoS Neglect. Trop. Dis.). Moreover, the CDC even uses "Growth in 0% NaCl" (see <https://www.cdc.gov/cholera/pdf/laboratory-methods-for-the-diagnosis-of-vibrio-cholerae-chapter-6.pdf>) as a diagnostic criterion for *V cholerae*.

Please also note that we demonstrated in our previous study, using three different methods (growth reporter, photobleaching, and two color ratio predictions upon growth/non-growth) that *V cholerae* bacteria grow very efficiently inside this vacuole. Thus, whatever the osmolarity is, *V cholerae* is able to very efficiently grow inside this organelle (Van der Henst et al., 2016).

Concerning the "hyperosmolar environment": Please note that all our experiments are done in half concentrated artificial seawater to mimic the brackish water environment in which *V cholerae* is often encountered. Notably, Cordingley et al. showed that high osmolarity stress triggers *Acanthamoeba castellanii* differentiation (=encystment) (Cordingley et al., 1996, J Cell Biochem). However, in our experiments, most of the amoebae do not encyst at around 30h p.p.c. if they were grown in the presence of a good bacterial food source (such as *E. coli* in the figure below) despite being in the half concentrated seawater medium. The absence of encystment under such conditions therefore suggests that the osmolarity does not lead to osmotic stress of *A. castellanii*.

Figure: Encystment of *A. castellanii*. Amoebae were cultures for 27h in half concentrated artificial seawater medium in the absence (-; no food) or presence (*E. coli*) of a food source.

Concerning "How does the hyperosmolar environment under which these infections are performed affect the conditions within the CV?" As mentioned above, the amoebae do not undergo osmotic stress-related encystment. Moreover, in time-lapse microscopy movies, we can observe that CV does periodically contract (e.g., for example in Movie S1 of Van der Henst et al., 2016). This is also the case after initial colonization by *V cholerae*. However, at a certain point the vacuole seems too tightly packed by matrix-embedded *V cholerae* (e.g., VPS, *Vibrio* polysaccharide), which is most likely why the vacuole "explodes" thereby releasing the bacteria into the cyst's cytosol.

14)

b) The life cycle of the amoebae should be described. Is encystation a baseline process or is it the result of infection in the high salt medium used for infection. Does *V. cholerae* alter the kinetics of encystation?

>> **Authors' comment:** In the revised manuscript, we now describe the biphasic life cycle of the amoebae in the introduction. Under the experimental conditions, most of the amoebae do not encyst if proper bacterial prey (=food) is present (see Figure above). However, in our previous study we showed that *V cholerae* is partly indigestible for the amoebae and exocytosed, indicating that it is not a good food source for the amoeba. Therefore, amoeba! encystation occurs at later time points (in contrast to *E. coli*) probably due to starvation of the amoeba (as also uninfected amoebae encysts, as can be seen for example in Fig. 3). Currently, we have no direct evidence that *V cholerae* directly triggers encystation though we do not exclude that the crowding in the contractile vacuole interferes with the osmoregulation of the amoebae and therefore accelerates encystation. Unfortunately, it is impossible for us to address the encystment kinetics due to technical limitations of our imaging approach (e.g., as movement of the amoebae does not allow us to follow single cells over very long times, as they often move out of the field of view).

15)

c) The findings of the previous manuscript should be discussed more clearly along with the open questions that are these were addressed here.

>> **Authors' comment:** The previous findings, namely that *V cholerae* can resist digestion/is exocytosed and, more importantly, can colonize the contractile vacuole and actively grow inside this organelle has been added to the introduction. The aim of this study was to study the underlying molecular mechanisms, which is stated in the introduction.

16)

d) The focus of the current introduction on the role of these minor virulence factors in mammalian hosts might be better saved for the discussion, where it can be compared with their role in the amoeba.

>> **Authors' comment:** We thank the reviewer for the comment. However, we kindly disagree, as we strongly believe that the introduction of these minor factors and their activities (e.g., hemolysin as a pore-forming toxin and the activity of the lecithinase) is crucial to understand the rationale of the result section. We therefore kept the structure of the manuscript in its original form.

17)

2) Even after reading both the published and current manuscripts, I am confused as to how *V. cholerae* accesses the CV. Is it creating a new pathway or co-opting an existing one?

>> **Authors' comment:** We apologize that this part wasn't clear. Indeed, a whole figure was dedicated to this process in our previous publication (Figure 3 and corresponding movies in Van der Henst *et al.*, 2016). Briefly, in this previous study, we demonstrated that the bacteria enter this compartment through vacuolar fusion of small food vacuoles with the contractile vacuole (panel B in this figure), which doesn't occur for control bacteria (e.g., *E. coli*) or fluorescent beads. As we had only briefly mentioned this fact in the initial manuscript, we now extended this section in the introduction in the revised manuscript to clarify this point.

18)

3) Please comment on the proportion of amoeba that become infected with *V. cholerae*. The requirement for the CLEM technique suggests that this is a very low frequency event.

>> **Authors' comment:** This assumption is exactly correct. The infection occurs at low frequencies but in a highly reproducible manner as previously discussed (Van der Henst *et al.*, 2016). And indeed, this is the reason why we chose the CLEM technique.

19)

4) Amoeba infected with HapA- strains are noted to have aberrant contractile vacuoles and do not encyst. If I understand this correctly, the authors conclude that the amoeba do not encyst because they are not viable. Is a direct test of the viability of HapA-infected amoebae possible? For instance, are markers of apoptosis present or is the cell membrane permeability increased?

>> **Authors' comment:** Thanks for this comment. As we shown in Fig. 2C, the cell membrane is not permeabilized, as the dextran doesn't enter into the cytosol. Unfortunately, most live-dead stains that we have tested do not work properly for *A. castellanii*. That's why we do not claim at this point that the amoebae are dead. However, these amoebae stop grazing and are floating in the device, as mentioned in the text. Thus, together with the impaired membrane morphology of the CV and the loss of tension of the latter, the cells are destined to die. The problem with long-term imaging is that these aberrant cells frequently float out of the field of view, which makes it technically impossible to follow their long-term fate.

20)

5) Line 127: Please define the term dissector pairs and give units if any for the value 1.35.

>> **Authors' comment:** We appreciate that the reviewer pointed out that this was a too EM-specific jargon. We therefore rephrased the paragraph to remove the disector pair term. As general information: A disector pair is two images a certain distance apart across which the count of bacteria is made.

21)

6) Line 258: This observation suggests that HapA results in complete degradation of GFP rather than just cleavage of HlyA with preservation of fluorescence. Is this the case? Western analysis would confirm complete degradation of the fusion protein.

7) Line 260: GFP alleles are notorious for aggregation, which results in mislocalization within cells. Because there is no evidence that this distribution of GFP is not an artifact, the authors should be more circumspect.

>> **Authors' comment:** We appreciate this comment. The observation suggests that either GFP is completely degraded or that the partial degradation of the fusion construct doesn't allow proper folding and/or maturation of GFP outside the bacterial cells. While western blotting might tell us about the degradation of the protein, the folding state cannot be judged using this method. We initially included the image of the HlyA-sfGFP fusion protein localization in the supplemental material, as we thought it showed a striking pattern that could hint at its localization inside the membrane. We truly believed that we mentioned this with precaution (e.g., "Interestingly, the sfGFP signal was not uniformly distributed within the contractile vacuole **but seemingly localized** to the edge in a patchy manner (Fig. S5b), **suggesting** oligomerization and **potential** pore formation within the membrane."). However, based on the reviewer's comment, we realized that this might indeed be too speculative. We therefore deleted this supplemental figure from the manuscript, as the data do not provide essential new information apart from what we anyway show throughout the manuscript using other methods (e.g., that the hemolysin intoxicates the amoeba).

22)

8) Line 330: Transcription of *hapA* is activated at high cell density and that *ohlyA* is repressed. At the same time, HapA degrades HlyA. This suggests that HlyA might play a role early in infection before high cell density is reached. Here, however, HlyA does not appear to be required for infection. How does the regulation of these two proteins fit in with the observations presented here and what might the role of HlyA be prior to activation of HapA?

>> **Authors' comment:** We fully agree with the reviewer that HlyA must have an additional role under so far unknown environmental conditions, especially given the enhanced production/activity of the protein in non-pandemic strains (as discussed above). However, here we focus on the interaction of *V. cholerae* with *A.*

castellanii and the bacterium's ability to use the CV and later on the cyst cytosol as a replication niche. Finding the primary role of this protein seems therefore beyond the scope of this study.

24)

9) Line 348: The authors have previously noted that escape from the CV is dependent on VPS. While this might be the result of osmotic pressure generated by VPS, more must be known about the physicochemical characteristics of the CV and the cytoplasm to evaluate this possibility. Furthermore, a defect in exopolysaccharide production has been shown to impact the transcription of many genes. Therefore, another possibility is that deletion of the *vps* genes decreases expression of factors required for CV escape.

>> **Authors' comment:** These are again excellent points. Notably, the current study does not directly focus on the effect of VPS. The essentiality of VPS to escape from the contractile vacuole has already been published before (Van der Henst *et al.*, 2016) and is therefore only discussed in the current manuscript, due to a recent supporting study by the Bassler and Wingreen labs (Yan *et al.*, 2017; *Nat. Comm.*; "Extracellular-matrix-mediated osmotic pressure drives *Vibrio cholerae* biofilm expansion and cheater exclusion"). Indeed, these authors wrote: "Recently, a new intracellular niche was discovered for *V. cholerae* in the marine amoeba *Acanthamoeba castellanii* (Van der Henst *et al.*, 2016). Interestingly, *V. cholerae* cells survive and replicate in this host's contractile vacuole, a dynamic organelle involved in the osmoregulation of the amoeba. Following intracellular replication, wild-type *V. cholerae*, but not EPS mutants, can destroy this organ and disperse from the host. We speculate that the capacity of the wild type to successfully disseminate from the vacuole could be related to the osmotic pressure response of the biofilm matrix shown here". We truly believe that the study by Yan *et al.*, 2017 strongly supports our previous observation on the essentiality of the VPS for *V. cholerae*'s escape from the CV and therefore kept this information in the discussion. However, we added the possibility that the accumulated VPS might solely impair the contractility of the CV, which would result in the same outcome (e.g., the lysis of the vacuole).

Concerning the second part of the comment "Furthermore, a defect in exopolysaccharide production has been shown to impact the transcription of many genes. Therefore, another possibility is that deletion of the *vps* genes decreases expression of factors required for CV escape". This is once again an interesting point. Unfortunately, we are not aware of any study in which the deletion of a *vps* gene that does not encode a regulatory protein but an enzyme that is involved in the VPS biosynthesis leads to transcriptional changes of many other genes. It is true that *vpsT* and *vpsR* mutants show vastly changed expression patterns, as both of these genes encode transcriptional regulators (e.g., Yildiz *et al.*, 2001, *J Bacteriol.* for VpsR; Casper-Lindley and Yildiz 2004, *J Bacteriol.* for VpsT; Beyhan *et al.* 2007, *J Bacteriol.* for a comparison between the VpsT and VpsR regulons).

However, to still address this concern, we performed motility experiments to check for changed motility of the *vpsA* mutant, as this mutant showed a lack of CV lysis in the previous publication (Van der Henst *et al.*, 2016). We also performed expression analyses on a set of diverse genes (e.g., those related to housekeeping functions, or phenotypes observed in this study, as well as known quorum-sensing related genes, etc). The data are shown below. None of this data points to any significant change in the *vpsA* mutant compared to the wild-type parental strain that would support the idea that the *vpsA* mutant is defective for CV lysis due to a secondary effect based on a changed expression pattern and not due to its primary phenotype, namely the lack of VPS biosynthesis.

C

Figure: Analysis WT versus *vpsA* mutant. (A) Representative image of motility plates of both strains. (B) Quantification based on three biologically independent experiments with two technical replicates each. Error bars represent SD. (C) Expression analyses comparing the WT strain (A1552) to its *vpsA*-minus derivative. Both strains were grown to high cell density (for 6h in LB) and reached comparable optical densities at 600 nm (WT=2.3, *vpsA*=2.2), which suggest normal growth of the mutant strain. The expression of representative genes was tested, which belong to different categories (color code from left to right): housekeeping genes including those encoding recombinases, chaperones, single-stranded binding protein, outer membrane porins, and catalase (green); cell wall and morphology genes (*O*-antigen (*wbe*) and cell shape (*crvA*) determining; red); genes encoding proteins involved in secondary messenger regulatory pathways (purple); genes encoding proteins that relevant to the current study (blue); biofilm-related genes (regulators and those encoding structural components of the matrix; yellow); another extracellular protease PrtV (light brown); type VI secretion genes (orange); and virulence genes (encoding toxin, structural component of the toxin-coregulated pilus, and major virulence regulators; turquoise). For none of these tested genes did we detect any significant difference in expression for the *vpsA*-minus mutant compared to WT.

25)

10) Where cell morphology was assessed, please comment in the methods section on whether the 6,000 cells counted were assessed manually or using a morphological analysis program.

>> **Authors' comment:** The amoebae were indeed assessed manually. We added this information to the methods section.

REVIEWERS' COMMENTS:

Reviewer #1 (Remarks to the Author):

The authors have satisfactorily addressed our concerns. We appreciate the authors' efforts in trying out the destabilized GFP versions. Not to their fault, this experiment failed. We agree with the author's assessment that spatiotemporal expression patterns is not essential to the main discovery of this communication. We hope that our comments have been otherwise helpful.

Reviewer #2 (Remarks to the Author):

In the revised manuscript, the authors have quite thoroughly responded to both reviewers' comments. I find the introduction much improved. Regarding the rebuttal, *V. cholerae* is reported to have a salt requirement, albeit much lower than that of other *Vibrios*, which is not incompatible with growth in some bodies of fresh water. Of course, it is also possible and even likely that *V. cholerae* modifies the vacuolar environment.

I have only one minor comment:

line 97: A clear explanation follows regarding expansion of the contractile vacuole (line 98). Therefore, the reference to "excess" water, which is vague, could be omitted.

Rebuttal letter 2

Molecular insights into *Vibrio cholerae*'s intra-amoebal host-pathogen interactions

>> Reviewer #1

The authors have satisfactorily addressed our concerns. We appreciate the authors' efforts in trying out the destabilized GFP versions. Not to their fault, this experiment failed. We agree with the author's assessment that spatiotemporal expression patterns is not essential to the main discovery of this communication. We hope that our comments have been otherwise helpful.

>> **Authors' comment:** We thank the reviewer for the kind words.

>> Reviewer #2

In the revised manuscript, the authors have quite thoroughly responded to both reviewers' comments. I find the introduction much improved. Regarding the rebuttal, *V. cholerae* is reported to have a salt requirement, albeit much lower than that of other *Vibrios*, which is not incompatible with growth in some bodies of fresh water. Of course, it is also possible and even likely that *V. cholerae* modifies the vacuolar environment.

I have only one minor comment:

line 97: A clear explanation follows regarding expansion of the contractile vacuole (line 98). Therefore, the reference to "excess" water, which is vague, could be omitted.

>> **Authors' comment:** We appreciate the reviewer's comment. However, we prefer to maintain this sentence in the text, as the reader should know that the discharged water is in excess. This information is important for the understanding of the overall study (e.g., how the amoebae behave if such excess water cannot be discharged anymore).